METHODS AND RESOURCES

# Cell2Spatial is a computational framework that maps single cells to spatial transcriptomic spots to reconstruct tissue architecture

Huamei Li[1], Jingchao Liu[2,3], Guige Wang[4], Zhenyu Liu[1], Meng Cao[5], Lingyun Sun[1]*, Cheng Peng[1]*, Yiyao Liu[6,7]*, Liang Ma[8]*, Qing Xiong [1]*

**1** Department of Rheumatology and Immunology, Nanjing Drum Tower Hospital, Affiliated Hospital of Medical School, Nanjing University, Nanjing, China, **2** TCRX (KeShiHua) Therapeutics Co. Ltd., Beijing, China, **3** Department of General Surgery, Sir Run-Run Shaw Hospital, Zhejiang University, Hangzhou, China, **4** Department of Thoracic Surgery, Peking Union Medical College Hospital, Beijing, China, **5** Department of General Surgery, Nanjing Drum Tower Hospital, Affiliated Hospital of Medical School, Nanjing University, Nanjing, China, **6** Department of Biophysics, School of Life Science and Technology, University of Electronic Science and Technology of China, Chengdu, China, **7** TCM Regulating Metabolic Diseases Key Laboratory of Sichuan Province, Hospital of Chengdu University of Traditional Chinese Medicine, Chengdu, China, **8** Department of Pharmacology, Biggs Institute for Alzheimer's and Neurodegenerative Disease, The University of Texas Health Science Center at San Antonio, San Antonio, Texas, United States of America

* lingyunsun@nju.edu.cn (LS); pengch@tcrximmune.cn (CP); liuyiyao@uestc.edu.cn (YL); mal1@uthscsa.edu (LM); qingx@tcrximmune.cn (QX)

## Abstract

Single-cell (SC) sequencing enables detailed characterization of transcriptional heterogeneity but lacks spatial context, while spatial transcriptomics (ST) preserves tissue organization yet is limited by resolution and incomplete gene capture. To bridge these gaps, we developed Cell2Spatial, a computational framework that segments spatial spots at single-cell resolution, even when SC and ST datasets are not fully matched in cell types. The method integrates information-theoretic gene selection, spatially weighted likelihood modeling, and spatial hotspot detection to improve signal fidelity. A corrected saturation model calibrates library size against gene complexity, ensuring accurate cell count estimation in low-resolution ST. To enhance scalability and spatial coherence, Cell2Spatial incorporates neural-network-guided clustering and a cost-minimizing assignment algorithm that balances transcriptional similarity with spatial proximity. Evaluations on synthetic data demonstrated that Cell2Spatial consistently outperforms existing tools in reconstructing tissue architectures and cellular compositions, with particular strength in handling unmatched datasets. Applications to 10× Visium data across mouse brain, human thymus, mouse kidney, and human dorsolateral prefrontal cortex revealed detailed anatomical structures and developmental trajectories. Moreover, for high-resolution platforms including Xenium In Situ, Visium HD, and Slide-seqV2, Cell2Spatial remained robust despite reduced transcript capture, effectively delineating fine-scale spatial patterns in complex tissues. Collectively, these results highlight Cell2Spatial as a versatile framework that

**Data availability statement:** All spatial transcriptomics datasets and their corresponding single-cell data were obtained from publicly available repositories, with detailed sources provided in S1 Table. Cell2Spatial, an R package designed to reconstruct tissue architecture at single-cell resolution by assigning single cells to spatial transcriptomics spots, is available on GitHub (https://github.com/lihuamei/Cell2Spatial) and Zenodo (DOI: https://doi.org/10.5281/zenodo.17217934). The full list of R packages used is specified in the Materials and methods section. Reproducible analysis code and the source data underlying the figures are openly accessible via GitHub (https://github.com/lihuamei/Cell2Spatial.Reproduce) and Zenodo (DOI: https://doi.org/10.5281/zenodo.17212677). The structured spatial and corresponding single-cell data can be accessed on Zenodo (DOI: https://doi.org/10.5281/zenodo.17430116).

**Funding:** This work was supported by grants from Jiangsu Funding Program for Excellent Postdoctoral Talent (2022ZB699 to H.M.L.), the National Key Research and Development Program of China (2020YFA0710800 to L.Y.S.), the Key Program of the National Natural Science Foundation of China (82330055 to L.Y.S.), the National Natural Science Foundation of China (12132004 and 32471367 to Y.Y.L.), the Alzheimer's Association and the National Alzheimer's Coordinating Center New Investigator Award (NIAP24_1273876 to L.M.), the Alzheimer's Association Research Grant—New to the Field (AARG-NTF-22-971669 to L.M.), and the NIH Award (P30AG072975 and K01AG084813 to L.M.). The funders had no role in study design, data collection and analysis, decision to publish, or preparation of the manuscript.

**Competing interests:** The authors have declared that no competing interests exist.

**Abbreviations:** CCA, canonical correlation analysis; cTECs, cortical thymic epithelial cells; DLPFC, dorsolateral prefrontal cortex; DN, double-negative; DP, double-positive; FNN, feedforward neural network; GSEA, gene set enrichment analysis; HVGs, highly variable genes; KL, Kullback–Leibler; mTECs, medullary thymic epithelial cells; PCA,

expands the analytical scope of ST and provides a powerful tool for uncovering the spatial organization of cellular function and tissue architecture.

## Introduction

Single-cell (SC) sequencing has evolved into an indispensable tool for conducting comprehensive investigations into gene expression and functional attributes within individual cells, thereby significantly advancing our understanding of the intrinsic heterogeneity among cell types and their pivotal roles in developmental and disease processes [1,2]. However, the process of SC sequencing inherently sacrifices spatial information during sample processing, a critical component essential for gaining insights into the dynamic cellular microenvironment [3]. Spatial transcriptomics (ST) technologies, like 10× Visium, preserve cell locations within tissue sections [4,5]. Nonetheless, ST technologies often grapple with challenges related to spatial resolution and the provision of comprehensive SC gene expression data. The integration of ST with SC data represents a highly effective approach for expediting the analysis of cell distribution within tissues, revealing intercellular relationships, and gaining a profound understanding of the structural and functional intricacies of tissues [6–9]. Regrettably, this integration encounters complexities stemming from the high-dimensional nature of the data.

To address this challenge, the widespread adoption of spatial deconvolution strategies has emerged. These strategies leverage SC reference data to deconvolute ST data and estimate the cellular composition within each spatial spot, as demonstrated by methods like Cell2location [10], CARD [11], RCTD [12], SpatialDWLS [13], STRIDE [14], DestVI [15], and Stereoscope [16]. However, these approaches are still constrained by resolution and typically provide information on the relative proportions of cell types, lacking detailed insights at the SC level, which hinders the in-depth exploration of cell states, interaction patterns, and neighboring cell populations. Establishing a channel that maps individual cells to specific spatial spots, enabling segmentation at SC granularity and potentially inferring unmeasured gene expressions, offers a broader perspective for the integrated analysis of ST and SC data [17]. Several methods have been publicly disclosed. One common approach, like Seurat [18], employs a label transfer method that maps SC labels onto spots. This method is based on Seurat's anchor-based data integration framework, which identifies shared features between the reference SC dataset and the query ST dataset, thereby enabling the transfer of cell type labels from SC to ST spots. Tangram [19] employs a deep learning framework to reveal intricate spatial structures at the SC level, aiding in the detection of complex cell distributions and relationships. On the other hand, CellTrek [3] utilizes a random forest model trained on ST data to predict spatial coordinates, benefiting from shared dimension reduction features with SC data, enhancing its efficiency with large-scale ST datasets. CytoSPACE [20] employs a correlation-driven cost function based on convex optimization to allocate individual cells to precise spatial spots, ensuring effective and reliable cell positioning.

principal component analysis; PCCs, Pearson correlation coefficients; RCC, renal cell cancer; RLM, robust linear regression model; RMSE, root mean square error; TLS, tertiary lymphoid structures; SC, single-cell; SGMA, Score-Guided Mapping Accuracy; ST, spatial transcriptomics; UMAP, Uniform Manifold Approximation and Projection; UMIs, unique molecular identifiers.

Despite these advances, current mapping tools face notable limitations. Label transfer-based strategies, such as Seurat's, struggle under certain conditions. For high-resolution (SC or subcellular resolution; e.g., image-based platforms) ST data, they can annotate cell types per spot while preserving the original gene expression profiles, which benefits mechanistic analyses. However, in platforms such as Xenium, the low transcript capture rate leads to extremely sparse expression matrices, limiting transcriptome coverage and reducing analytical depth. For low-resolution (multiple cells per spot; e.g., spatial barcoding-based platforms) ST data, these tools typically assume each spot originates from a single cell type, overlooking potential cell mixtures within spots and thereby introducing bias. Beyond label transfer, SC mapping strategies also encounter two major challenges. First, aligning unpaired SC and ST datasets is difficult, particularly in minimizing mismatches and missing mappings. Second, in low-resolution ST data, spots frequently contain multiple cells, complicating estimates of cell counts and the inference of spatial relationships within heterogeneous neighborhoods. Different tools address these challenges with varying trade-offs. CellTrek [3] performs well in mapping unpaired SC and ST datasets, assigning cell types to corresponding ST regions, but it has limited ability to resolve cellular composition or fine-grained spatial relationships within spots. Conversely, CytoSPACE [20], by integrating the RCTD framework [12], provides more accurate estimates of cellular composition and spatial distribution, yet it assigns cells at the tissue-wide level and does not resolve the issue of unpaired SC and ST mapping. Together, these limitations underscore the need for new methods that can simultaneously handle unpaired SC–ST data integration and accurately infer spatial relationships within heterogeneous spots—an essential step toward advancing integrative SC and ST research.

Herein, we present Cell2Spatial, a unified framework for reconstructing tissue architecture at SC resolution. Cell2Spatial integrates a spatially weighted maximum likelihood model with spatial hotspot detection to robustly quantify single-cell spot similarity and reduce mismatches between SC and ST datasets. For low-resolution platforms, it employs a corrected saturation model to estimate spot-level cell counts by modeling the relationship between library size and gene counts. To enable efficient large-scale mapping, a feedforward neural network (FNN) is combined with a linear sum assignment algorithm, ensuring accurate and scalable cell-to-spot allocation. Compared with existing approaches, Cell2Spatial offers a practical improvement in reconstructing tissue structure and cellular composition, seamlessly integrates unpaired SC–ST data, and reliably captures the spatiotemporal dynamics of cell states.

## Results

### Overview of Cell2Spatial

The Cell2Spatial framework aims to map single cells to spatial spots for reconstructing tissue architecture (Fig 1). The process begins with normalizing SC and ST data, ideally from the same tissue source, using Seurat's *SCTransform* to establish a common basis for gene expression comparison. To quantify similarities between

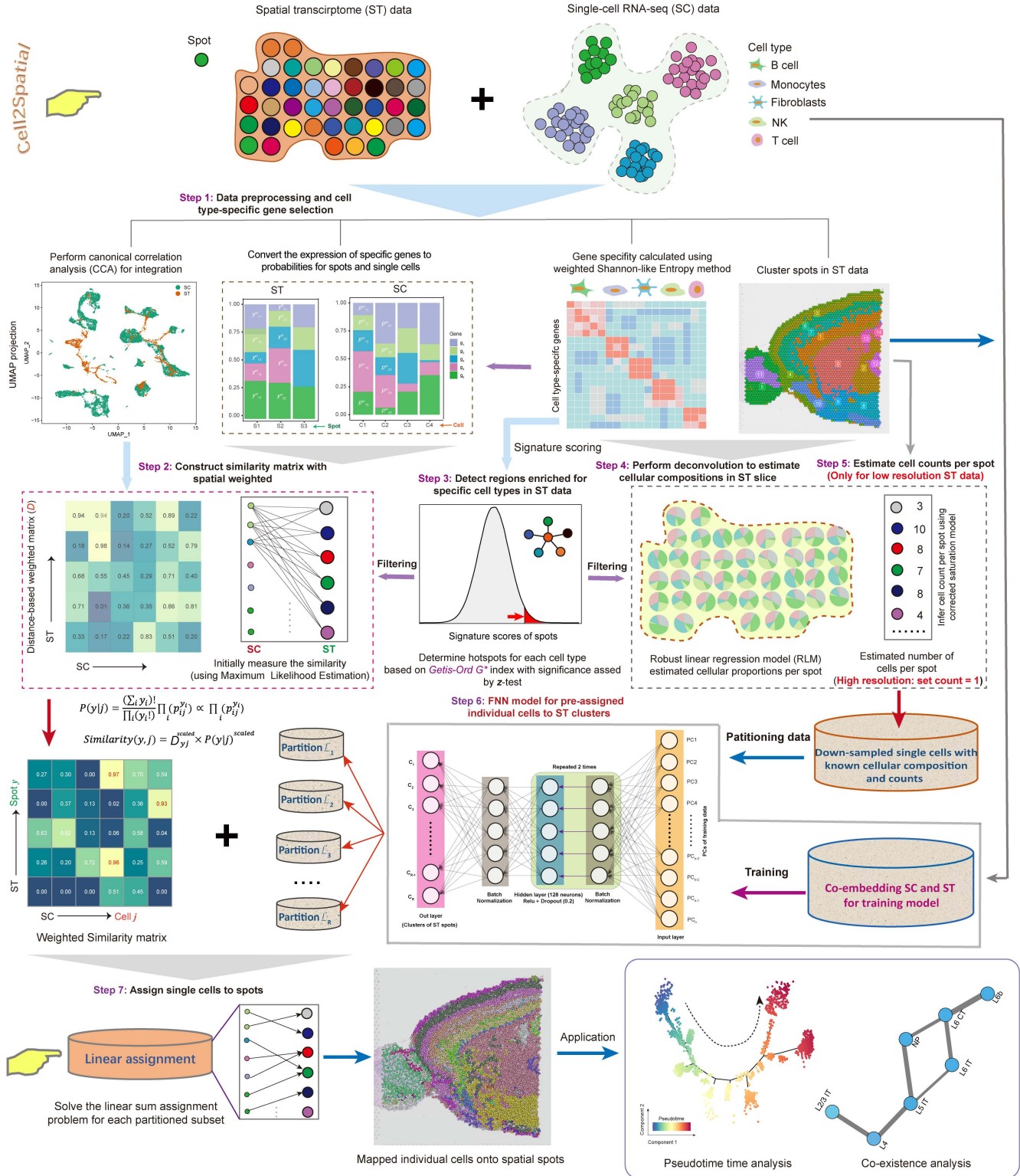

**Fig 1. Overview of the Cell2Spatial algorithm.** The Cell2Spatial framework comprises seven fundamental components: data preprocessing, marker gene selection, similarity matrix construction, hotspot detection, deconvolution and cell count estimation, single-cell (SC) pre-assignment, and final mapping to spots (see "Materials and methods"). SC and spatial transcriptomics (ST) data were standardized using *SCTransform* in Seurat, with

cell-type-specific genes identified through a modified entropy-based method (Step 1). A spatially weighted similarity matrix was created using a maximum likelihood model, with Seurat's canonical correlation analysis (CCA) connecting single cells to spatial spots to yield spatial weights (Step 2). Cell type hotspots were identified using the Getis-Ord G* index combined with normality testing, while robust linear regression was applied for deconvolution of cellular composition, and corrected saturation model was used to estimate cell counts in low-resolution ST data (Steps 3-5). Feedforward neural network (FNN) was employed for cell pre-assignment to ST clusters (Step 6), and the Jonker-Volgenant algorithm iteratively mapped individual cells to specific spatial spots based on similarities and estimated cell counts (Step 7).

single cells and spatial spots, maximum likelihood model is employed. Efficiency is enhanced by identifying cell-type-specific genes through a modified entropy-based method (Fig 1, Step 1; see "Materials and methods"). These gene sets, along with their expression profiles, form probability matrices that construct the maximum likelihood model and generate a likelihood matrix. Spatial context is incorporated through Seurat's canonical correlation analysis (CCA) [18] for SC–ST integration, followed by principal component analysis (PCA) for dimensional reduction. A Euclidean distance matrix is derived from the Uniform Manifold Approximation and Projection (UMAP) embedding, constructed on the PCA space. This distance matrix is applied as weights (scaled to 1) to the likelihood matrix, resulting in a weighted likelihood matrix that balances gene expression similarity with spatial proximity (Fig 1, Step 2; see "Materials and methods").

To improve robustness, particularly when SC and ST data may not perfectly align in terms of cell types, the weighted likelihood matrix undergoes refinement. Spots that do not correspond to any pre-defined cell types in the SC dataset are filtered out determined by the Getis-Ord *G** index coupled with a normal test based on signature scores from cell-type-specific genes, ensuring focus on relevant spatial regions (hotspot regions) while minimizing mapping noise (Fig 1, Step 3; see "Materials and methods"). Understanding the cellular compositions within ST sections is critical for recovering tissue spatial organization. This is achieved through a deconvolution model that applies robust linear regression model (RLM) in conjunction with base matrix constructed from specific gene sets (Fig 1, Step 4; see "Materials and methods") [21,22]. The corrected saturation model, was designed to estimate cell counts per spot in low-resolution ST data by linking the library size and gene count, through a saturation-based relationship (Fig 1, Step 5; see "Materials and methods"). For high-resolution ST data, where spots typically capture single cells, it assigns a cell count of 1 per spot. Therefore, the total number of cells to be aligned is calculated as the sum of cell counts across the remaining spots, with the number of cells for each type sampled from the SC dataset based on these totals and the estimated cellular proportions, yielding candidate mapping SC data.

To assign candidate single cells to spatial spots, the Jonker-Volgenant algorithm [23] is employed, utilizing the refined weighted likelihood matrix and the estimated cell counts per spot. In large-scale applications, however, this algorithm imposes substantial memory and processing demands. To mitigate these challenges, a FNN framework is implemented. The FNN is trained to pre-allocate cells to ST clusters based on the co-embedding of SC and ST data (Fig 1, Step 7; see "Materials and methods"). This pre-allocation streamlines the execution of the Jonker-Volgenant algorithm, allowing it to be applied iteratively within each ST cluster, ensuring efficient and scalable SC-to-spot assignments (Fig 1, Step 7; see "Materials and methods").

### Evaluation of the performance of Cell2Spatial using mouse brain transcriptome data

To evaluate the performance of Cell2Spatial, ST data from the 10× Visium platform for the mouse brain was obtained, comprising 2,696 spots. Among these, around 66.53% of genes exhibited no expression, with the highest UMI count recorded at 25,580 and a median count of 24,332 UMIs (S1A Fig and S1 Table). Additionally, we utilized a reference dataset (SC) from the Allen Institute, which comprises approximately 14,000 adult mouse cortical cells, covering 23 distinct cell types (S1B Fig). As the Fig 2A shown, the application of Cell2Spatial enabled us to map the single cells to precise spatial spots, and the findings unveiled the presence of distinct anatomical distributions of various cell types within the mouse

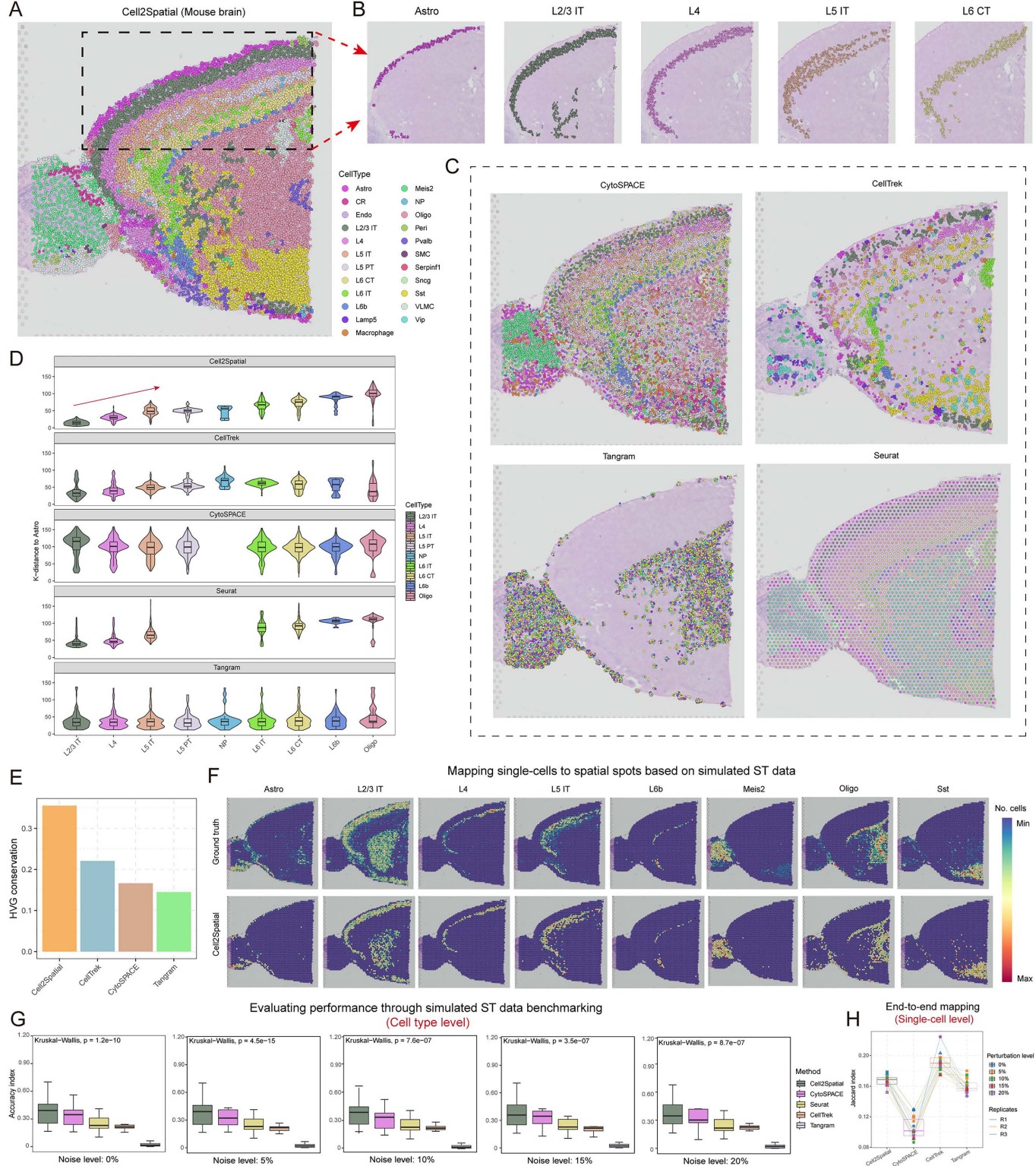

**Fig 2. Performance assessment of Cell2Spatial on real and simulated mouse brain spatial transcriptomics (ST) data. (A)** Spatial architecture of mouse brain reconstructed using Cell2Spatial. Cell types are marked by color codes, and each dot represents an individual cell. **(B)** Distribution of cell types Astro, L2/3 IT, L4, L5 IT, and L6 CT on mouse brain ST slice. **(C)** Spatial architecture of the mouse brain reconstructed using CytoSPACE,

CellTrek, Tangram, and Seurat, respectively. Cell types are color-coded. Each dot represents an individual cell. **(D)** Spatial *k*-distance ($k = 10$) of L2/3 IT, L4, L5 IT, L5 PT, NP, L6 IT, L6 CT, L6b, Oligo to Astro cells. Violin boxplots illustrate the median, and interquartile ranges (25%–75%), with whiskers extending up to 1.5 times the interquartile range beyond the box. **(E)** Bar plot showing the conservation of highly variable genes (HVGs) between reconstructed spatial expression data from mapping tools and the original spatial expression data (see "Materials and methods"). (F) Spatial heat maps showing the performance of Cell2Spatial for aligning scRNA-seq data (with 5% added noise) to spatial spots in ST datasets simulated with five cells on average (see "Materials and methods"). Only cell types with prominent spatial structures have been presented for the sake of clarity. The color intensity of each spot corresponds to the number of mapped single cells. **(G)** Box plots showing the performance across various methods, and noise levels (0, 5%, 10%, 15%, and 20%) in accurately assigning individual cells to their respective locations in simulated ST datasets. Each data point represents a distinct cell type. The central lines within the boxes, the box edges, and the whiskers correspond to the medians, the first and third quartiles, and the minimum and maximum values, respectively, within 1.5 times the interquartile range of the box boundaries. *P*-values were obtained by Kruskal–Wallis test, as it is appropriate for comparing more than three groups of non-normally distributed data without assuming equal variances. **(H)** Boxplot showing the end-to-end Jaccard index at the single-cell level, measuring the consistency between the coordinates of single cells in the reconstructed data and their coordinates in synthetic ST data. Perturbations were introduced by randomly modifying gene expression in 0%, 5%, 10%, 15%, or 20% of genes. For each perturbation level, three independent replicates were generated to ensure robustness of the evaluation. The data underlying this figure can be found at https://doi.org/10.5281/zenodo.17212677.

brain (Fig 2A and 2B). Notably, our approach capitalizes on a modified specificity entropy strategy for the identification of cell-type-specific genes, showcasing noteworthy discriminatory power and demonstrating its effectiveness (S1C Fig).

Next, the performance of Cell2Spatial with other mapping tools capable of segmenting spatial spots into SC granularity was compared, including CytoSPACE [20], CellTrek [3], Tangram [19], and Seurat [18]. The findings indicated that, the performance of the other four mapping tools within the ST sections was significantly weaker than that achieved by Cell2Spatial (Figs 2C and S1D–S1G). Importantly, Tangram's mapping did not reveal a distinct alignment between cell types and anatomical structures, and Seurat directly assigned spots as individual cell types. Furthermore, as shown in Fig 2D, Cell2Spatial generated consistent spatial relationships between specific cell types (L2/3 IT, L4, L5 IT, L5 PT, NP, L6 IT, L6 CT, L6b, Oligo, to Astro) based on *k*-distances, which were in agreement with previous studies [3,19] (Fig 2D and S2 Table). CellTrek was also efficient, whereas Tangram exhibited less favorable performance. Additionally, we reconstructed spot-level expression profiles using the mapped SC resolution ST dataset and assessed highly variable genes (HVGs) conservation, with Cell2Spatial demonstrating the highest conservation compared to unprocessed ST data (Fig 2E; Cell2Spatial: 0.357, CytoSPACE: 0.166, CellTrek: 0.221, Tangram: 0.145; see "Materials and methods"). This indicates its effectiveness in preserving the intrinsic gene characteristics of spatial spots within tissue sections. Notably, since Seurat's label transfer function was used, the spot-level expression in the ST data remained unchanged post-transfer and was excluded from this analysis.

Nevertheless, the limited information regarding cell composition and location within the real ST data of mouse brain mitigate our evaluation of Cell2Spatial's performance. To overcome this challenge, simulated ST datasets of the mouse brain were generated with accurately known cell type localization and composition at various noise levels (see "Materials and methods"). Applied Cell2Spatial, CytoSPACE, CellTrek, Tangram, and Seurat to map the SC data of the mouse brain to the simulated ST datasets, our results demonstrated that Cell2Spatial adeptly restored the spatial positions of cell types, and exhibited a strong alignment with actual cell counts of spots, as reflected by the root mean square error (RMSE) between predicted and expected counts (Cell2Spatial: 2.21; CytoSPACE: 3.29; CellTrek: 4.10; Tangram: 8.52; Seurat: 6.38) (Figs 2F and S2A–2C). We simulated spots with varying cell counts to further evaluate Cell2Spatial's ability to estimate cell numbers per spot (S2D Fig). Across different values of *Lambda* (*Lambda* = 1:20; representing the Poisson distribution parameter used to generate random cell counts), Cell2Spatial consistently achieved Pearson correlation coefficients (PCCs) above 0.89 (median: 0.941; maximum: 0.952), whereas CytoSPACE achieved coefficients above 0.61 (median: 0.693; maximum: 0.783), demonstrating a significant performance difference (Two-sided Wilcoxon test; *p*-value < 0.001) (S2E–S2G Fig and S3 Table). Consistently, RMSEs were also significantly lower for Cell2Spatial (median: 1.15) compared to CytoSPACE (median: 2.76) (Two-sided Wilcoxon test; *p*-value < 0.001) (S2G Fig).

To assess performance more effectively, we simulated ST datasets with known cell type composition and spatial distributions under different perturbation levels (see "Materials and methods"). Based on accuracy indices, Cell2Spatial consistently outperformed other tools at the cell type level across all noise conditions (Fig 2G and S4 Table). Additionally, we investigated a crucial aspect of the analysis: mapping SC datasets designed for simulating ST data to their corresponding synthetic ST data to assess the accuracy of cell localization—i.e., end-to-end mapping. To minimize the influence of stochastic processes on performance evaluation, we performed three replicates under each perturbation level. The results indicated that CellTrek achieved the highest Jaccard index (mean index: 0.193), with Cell2Spatial performing next (mean index: 0.167), outperforming both CytoSPACE (mean index: 0.106) and Tangram (mean index: 0.160) in overall accuracy (Fig 2H). The relatively strong accuracy of CellTrek may stem from its Random Forests framework, which can better capture complex, non-linear relationships in ST data and provide greater robustness to noise compared with the linear assignment algorithm used by Cell2Spatial. Notably, all tools showed low Jaccard indices (<0.20), reflecting a potential limitation: current methods mainly distinguish between cell types but lack power to resolve cells within the same type, making them better at reconstructing cell-type distributions than pinpointing individual cell locations.

Taken together, Cell2Spatial can accurately infer the composition of cell types within spatial structures, and effectively reconstruct spatial architectures for tissues.

## Assessing Cell2Spatial's effectiveness across diverse metrics and conditions

ST datasets with varying spot resolutions (*Lambda* = 5, 10, 15, 20) and cellular compositions were simulated, where *Lambda* served as the Poisson distribution parameter for randomly generating cell numbers per synthetic spot. We initially focused on Cell2Spatial's ability to predict cellular proportions within these simulated ST sections. To facilitate comparison, we incorporated 10 additionally spatial deconvolution tools: Cell2location [10], SpatialDWLS [13], RCTD [12], Stereoscope [16], DestVI [15], SpaOTsc [24], novoSpaRc [25], SPOTlight [26], CARD [11], and DSTG [27], using SC data from the mouse brain as the reference to estimate cellular proportions in each spot from the synthetic ST datasets. For mapping tools, including Cell2Spatial, CytoSPACE, CellTrek, and Tangram, we aligned single cells from mouse brain to the synthetic ST datasets, and then calculated the cellular proportions. In the case of Seurat's label-transfer method, we utilized the probability matrix obtained through the "*TransferData*" function to represent cellular proportions in each spot. As shown in Fig 3A, across various spot resolutions (*Lambda* = 5, 10, 15, and 20), Cell2Spatial (PCCs: 0.969–0.976; RMSEs: 0.0135–0.0148) outperformed similar tools such as CytoSPACE (PCCs: 0.910–0.974; RMSEs: 0.0134–0.0271), CellTrek (PCCs: 0.777–0.833; RMSEs: 0.0332–0.0402), and Tangram (PCCs: 0.246–0.354; RMSEs: 0.0541–0.0615) in predicting cellular proportions (Fig 3A; see "Materials and methods"). Additionally, when compared to the spatial deconvolution tool CARD, Cell2Spatial demonstrated comparable efficacy in predicting cellular compositions, achieving PCCs greater than 0.97, and RMSEs below 0.015 across all resolutions (Fig 3A; bottom panel). We further benchmarked Cell2Spatial using 32 published SC and simulated ST datasets from Li and colleagues [28], which cover human brain, liver, lung, kidney, and pancreas, as well as mouse pancreas and trachea. These datasets contain a median of eight cell types (range: 6–128), with eight datasets including more than 10 cell types. Each simulated dataset comprises 1,000 spots with known cellular compositions. On these benchmarks, Cell2Spatial demonstrated strong performance in reconstructing spot-level cell type compositions, achieving an average PCC of 0.784 (range: 0.427–0.964; highest in dataset30, lowest in dataset1), ranking second among all tools, just behind Cell2location (range: 0.619–0.942) (Fig 3B). For RMSE, Cell2Spatial achieved an average of 0.114 (range: 0.027–0.220), ranking fifth overall. While Cell2location (range: 0.031–0.145) showed the best overall accuracy, Cell2Spatial maintained competitive performance in estimating cell type compositions (Fig 3B). These benchmarking results on both simulated and published datasets confirm that Cell2Spatial achieves high accuracy and robustness across diverse conditions.

Next, to assess the spatial density of cell charting results against the actuary spatial distribution for various cell types in the synthetic ST datasets (Fig 3A), *Kullback–Leibler* (*KL*)-divergence was used. The analysis showed that Cell2Spatial

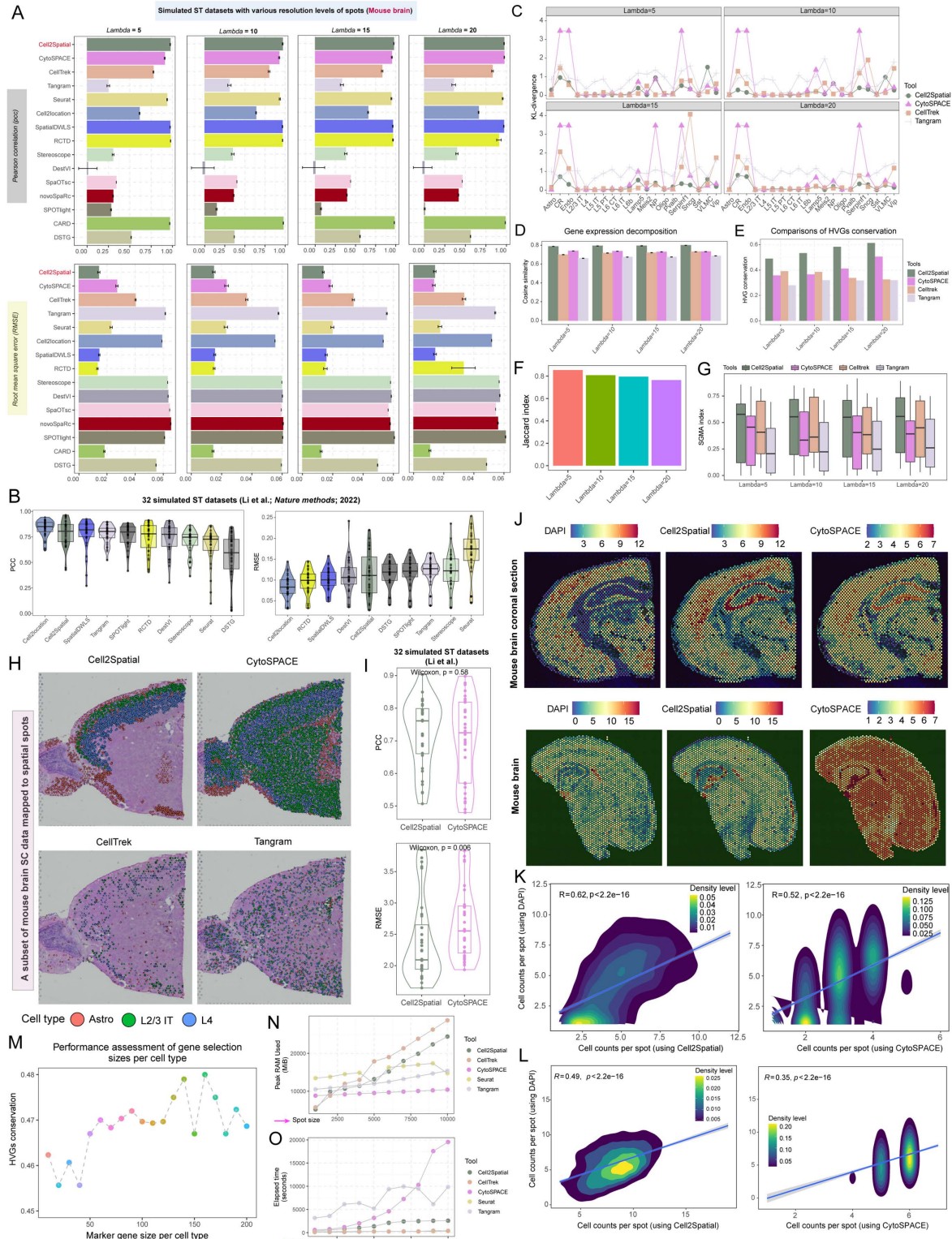

**Fig 3. Evaluation of spatial transcriptomics (ST) tools across various metrics and conditions. (A)** Bar plots showing Pearson correlation coefficients (PCCs) and Root Mean Square Errors (RMSEs) between predicted and true proportions in synthetic ST datasets under different spot resolution levels (*Lambda* = 5, 10, 15, and 20). The *Lambda* value represents the expected cell counts in spots, modeled by a Poisson distribution. Error bars

indicate variability, determined through a permutation strategy repeated 100 times, where 2,000 spots were down-sampled, and PCC and RMSE were calculated for each iteration. **(B)** Violin combined with box plots showing the PCCs and RMSEs between estimated and actual cellular proportions. Black lines within the boxes indicate medians. Thirty-two simulated ST datasets, together with their corresponding single-cell (SC) datasets, were retrieved from the study by Li and colleagues [28]. **(C)** Kullback–Leibler (*KL*) divergence metrics for spatial cell charting methods, comparing each cell type against synthetic ST dataset references. **(D)** Grouped bar plots showing cosine similarities between recovered spot expressions post-SC mapping and the original synthetic ST dataset expressions. Error bars represent values obtained via permutation strategy, with 2,000 spots down-sampled and cosine similarities calculated across 100 iterations. **(E)** Grouped bar plots showing the conservation of highly variable genes (HVGs) between recovered spot expressions and synthetic ST dataset expressions following SC mapping. **(F)** Bar plot showing the average Jaccard index for cell type consistency in spots, comparing predictions from our score-guided mapping accuracy (SGMA; see "Materials and methods") strategy against true cell types in synthetic ST datasets. **(G)** Box plot displaying SGMA indexes for cell type consistency in spots, comparing predictions using mapping tools (Cell2Spatial, Cyto-SPACE, CellTrek, and Tangram) with true cell types in synthetic ST datasets, based on the SGMA strategy (see "Materials and methods"). **(H)** Scatter plot mapping a subset of mouse brain SC data to ST spots under unmatched conditions (where cell types from SC data represent only a subset of those in ST datasets). Each dot corresponds to an individual cell, with cell types color-coded. **(I)** Violin combined with box plots showing the PCC and RMSE between estimated and actual cell counts in spots. Simulated ST datasets, together with their corresponding SC datasets, were retrieved from the study by Li and colleagues [28]. **(J)** Spatial feature plots illustrating inferred cell counts in spots using the DAPI channel in fluorescence images (counted via Squidpy [29]), Cell2Spatial, and CytoSPACE. The top panel represents a coronal section of the mouse brain, while the bottom panel represents the whole brain. Color shading indicates different cell counts in spots. **(K, L)** Density plots combined with fitted lines, showing the consistency between cell counts estimated by Cell2Spatial or CytoSPACE and those counted using DAPI via the Squidpy tool [29]. "*R*" represents the Spearman correlation, with *p*-values obtained from the two-sided *t*-tests. **(M)** Scatter plot with line showing the influence of marker size on Cell2Spatial performance for different cell types. The marker size range is 10 to 200, with a step size of 10. **(N, O)** Scatter plots with line showing peak memory usage and elapsed time for different mapping tools under varying spot sizes, ranging from 1,000 to 10,000, with a step size of 1,000. The data underlying this figure can be found at https://doi.org/10.5281/zenodo.17212677.

had consistently lower *KL*-divergence values than CytoSPACE, CellTrek, and Tangram, indicating that Cell2Spatial more accurately recovered spatial cellular structures (Fig 3C; see "Materials and methods"). We also evaluated gene expression recovery by comparing the cosine similarity between the recovered spot expressions and the original synthetic datasets. Cell2Spatial consistently achieved the highest similarity (approximately 0.8) across all datasets, highlighting its effectiveness in accurately reconstructing gene expression patterns (Fig 3D; Cell2Spatial: 0.787–0.798; CytoSPACE: 0.779–0.789; CellTrek: 0.753–0.778; Tangram: 0.712–0.736). Furthermore, we assessed HVGs conservation, which serves as a proxy for the preservation of the biological signal in the ST expression profile before and after mapping, and found that Cell2Spatial performed well in retaining HVGs across all spot resolutions, demonstrating its ability to preserve key gene expression features effectively (Fig 3E; Average HVGs conserved scores: Cell2Spatial (0.556); CytoSPACE (0.411); CellTrek (0.361); Tangram (0.310)). Moreover, we proposed a metric, the Score-Guided Mapping Accuracy (SGMA) index. This method scores spots using marker genes for specific cell types, capturing the prominent spatial regions of these cell types and comparing the consistency of these regions in the ST spatial data before and after mapping (see "Materials and methods"). Using marker genes to score synthetic ST spots, we identified high-confidence spots for each cell type. The results, as shown in Fig 3F, demonstrated that the cell type distributions inferred by SGMA strategy were highly consistent with the true distributions, with Jaccard indexes exceeding 0.75 (Fig 3F; range: 0.764–0.853), demonstrating SGMA's effectiveness in evaluating mapping performance. Applying SGMA index to various mapping tools across multiple synthetic ST datasets, we found that Cell2Spatial achieved the highest median Jaccard index for recovering ST cell type distributions, reinforcing its strong performance in cell type identification (Fig 3G; Cell2Spatial: 0.503–0.560; CytoSPACE: 0.289–0.497; CellTrek: 0.304–0.428; Tangram: 0.248–0.315). Notably, SGMA index is not only valuable for synthetic ST datasets but also serves as an effective benchmark for evaluating mapping tools on real experimental ST data. In particularly, to investigate the performance of mapping tools under mismatched conditions, where the cell types in the SC data represent only a subset of those in the ST dataset, we focused on three cell types—Astro, L2/3 IT, and L4—from the mouse brain SC dataset. We used four mapping tools to project single cells from these cell types to ST spots. The results revealed that Cell2Spatial effectively reconstructed the spatial distribution of these three cell types (Fig 3H). In contrast, CytoSPACE, which employs a full spatial slice mapping approach, struggled to accurately reconstruct the tissue's spatial

structure under these mismatched conditions. CellTrek partially captured the spatial distribution of the three cell types but was less precise for Astro cells, while Tangram did not clearly reveal the spatial distribution of these cell types. These findings highlight Cell2Spatial's capability to handle complex and mismatched datasets effectively.

To validate the accuracy of spot-level cell count estimation by Cell2Spatial, we applied both Cell2Spatial and CytoSPACE to 32 simulated ST datasets generated by Li and colleagues [28], each containing spots with known cell numbers. Cell2Spatial achieved higher correlations with true counts (PCC range: 0.507–0.903; mean = 0.725) compared to CytoSPACE (PCC range: 0.480–0.898; mean = 0.700) (Fig 3I). RMSE values further confirmed its advantage, with Cell2Spatial showing significantly lower errors (range: 1.643–3.846; mean = 2.390) relative to CytoSPACE (range: 1.935–3.834; mean = 2.683) (Fig 3I). We next applied Cell2Spatial to two mouse brain ST datasets (10× Visium) with available DAPI channel images. Cell counts were quantified from DAPI staining using Squidpy [29] and compared with estimates from Cell2Spatial and CytoSPACE. Cell2Spatial's estimates more closely matched the DAPI-based counts, achieving correlation coefficients of 0.62 and 0.49, outperforming CytoSPACE (coefficients: 0.52 and 0.35) (Fig 3J–3L; see "Materials and methods"). To assess the effect of predefined maximum cell numbers ($n^{max}$), we varied $n^{max}$ between 4 and 24 in a synthetic mouse brain ST dataset (true $n^{max} = 14$). As $n^{max}$ deviated from the true value, RMSEs increased (S2H Fig; left panel). Nonetheless, PCCs consistently exceeded 0.8, indicating that despite differences in $n^{max}$, the overall trend of spot-level cell distribution was preserved (S2H Fig; right panel). Considering the central role of cell type-specific genes in Cell2Spatial, we investigated how varying the number of selected specific genes per cell type affected Cell2Spatial's performance. The results revealed that while increasing the number of specific genes enhanced the conservation of HVGs in the reconstructed spatial expression profiles, Cell2Spatial's performance remained relatively stable within a certain range (HVGs conservation: 0.47 ± 0.15) (Fig 3M). This stability suggests that Cell2Spatial is resilient to changes in gene selection. Finally, we evaluated the computational efficiency of the mapping tools. Cell2Spatial demonstrated relatively high memory usage, though it stayed within acceptable limits, and consistently achieved relatively short runtimes even as the spot size increased (Fig 3N and 3O; see "Materials and methods"). In contrast, CytoSPACE and Tangram showed a linear increase in computational time, highlighting Cell2Spatial's good efficiency and practical advantages.

Overall, our evaluation shows that Cell2Spatial performs well across various metrics and conditions. Its effectiveness in estimating cellular compositions, recovering gene expression, and maintaining computational efficiency, along with its adaptability to different input parameters, makes Cell2Spatial a valuable tool for ST analysis.

## Spatial profiling of single-cells in various tissue types with Cell2Spatial

To assess the ability of Cell2Spatial in analyzing ST data from diverse tissue sources, ST data [30] along with a reference scRNA-Seq dataset of human thymus tissue [31] were obtained. The thymus ST dataset consisted of 1,628 spots, with a maximum of 38,161 unique molecular identifiers (UMIs) and a median of 91,66 UMIs, while the SC dataset encompassed 40 distinct cell types. Through the process of assigning cells to ST spots, Cell2Spatial successfully mapped 39 cell types to the thymus section, revealing notable differences in cell type distribution between the medulla and cortical regions (Figs 4A and S3A–S3D and S1 Table). As known, the human thymus is the site of T-cell development, with double-negative (DN) and double-positive (DP) T cells residing in the cortical region, while single-positive (SP) cells are mainly localized to the medullary region (Fig 4B). We tested the ability of various mapping tools to capture the thymic T cell development process, with focus on the DN, DP, ABT.entry (transition from DP to SP states), CD8+ T, CD4+ T cells, and cortical thymic epithelial cells (cTECs) and medullary thymic epithelial cells (mTECs) that are involved in T cell maturation. Cell2Spatial revealed distinct spatial patterns, with DN and DP cells predominantly inhabiting the thymic cortex, and ABT.entry cells strategically positioned in the cortical region near the corticomedullary junction (Fig 4C). On the other hand, CD8+ T and CD4+ T cells, representing mature SP cells, were predominantly found in the thymic medulla (Fig 4C). Furthermore, cTECs were primarily concentrated in the cortical region, while mTECs were detected in the medullary region, consistent with previous researches [31–33]. Interestingly, the presence of a small subset of cTECs in the medullary region hints

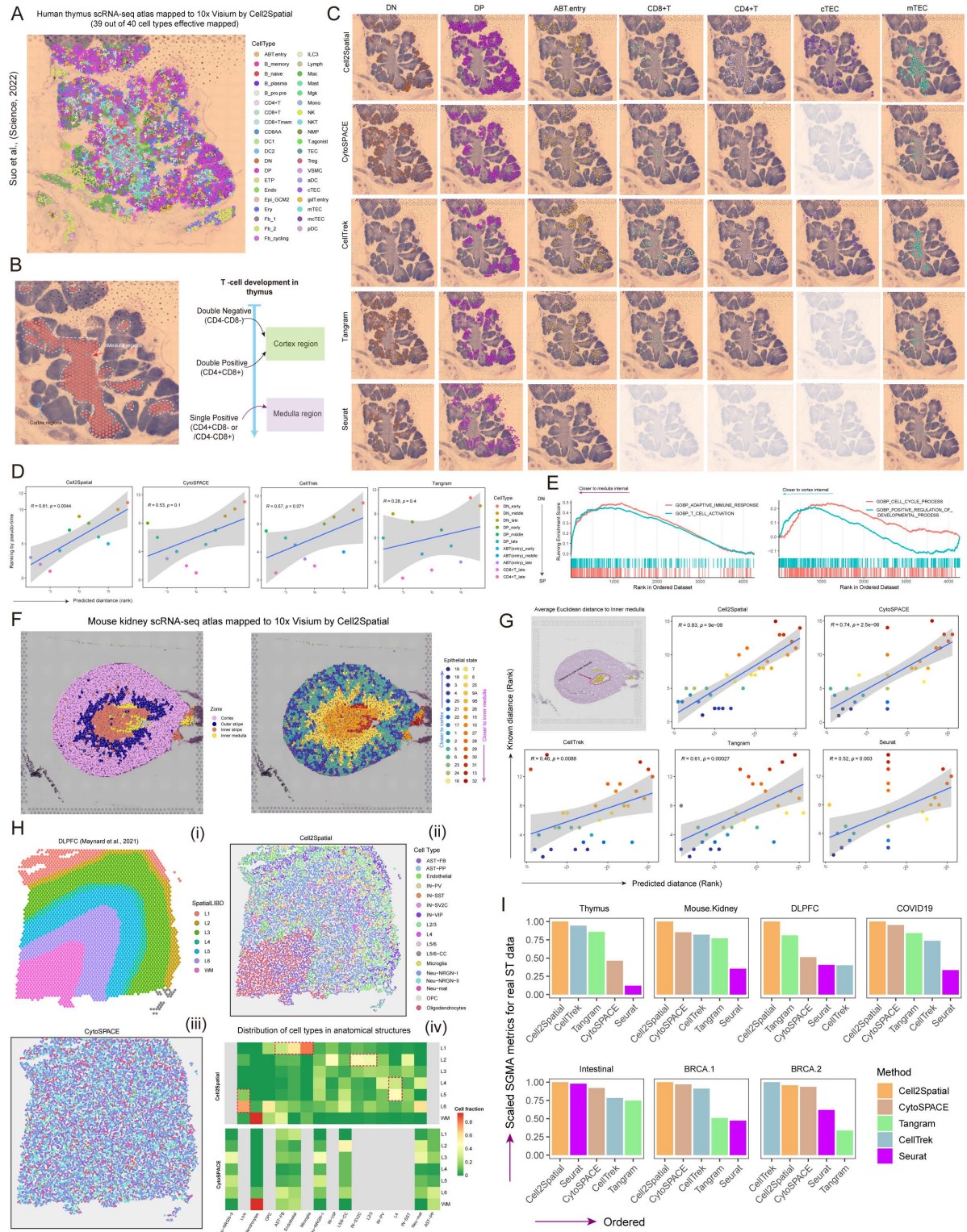

**Fig 4. Performance assessment of Cell2Spatial in real ST data from various tissue types. (A)** Spatial architecture of the human thymus (Suo and colleagues) [30] reconstructed using Cell2Spatial. Cell types are color-coded. Each dot represents an individual cell. **(B)** Thymus anatomy and the T cell development process. (Left) Hematoxylin and eosin (H&E) staining images depicting the thymus structure. The white line marks the medullary-cortical

junction, with colored regions indicating the medullary areas and uncolored regions representing the cortical areas. (Right) T cell development stages, encompassing the double-negative (DN), double-positive (DP), and single-positive (SP) stages. DN and DP stages were primarily located in the cortical regions, while the SP stage is concentrated in the medullary regions. **(C)** Distribution of selected cell types on human thymus ST slices. From left to right, cell types include DN, DP, ABT.entry, CD8+ T cells, CD4+ T cells, cTECs, and mTECs. **(D)** Scatter plots of various mapping tools showing the concordance between the relative Euclidean distance ranking of T cell subtypes at different stages and their developmental sequence determined by pseudo-time analysis. Euclidean distances of individual cells were calculated relative to a reference point at the center of the medullary region (regions 3, 7, and 8 in S3A Fig, left panel). For each subtype, the mean distance from the reference point was used to establish a spatial ranking, which was then compared with the developmental order. The strength of this association was quantified using Spearman correlation analysis. The blue line represents a linear fit, and the shaded area denotes 95% confidence interval. Different colored dots represent distinct T cell subgroups. "*R*" stands for the Pearson correlation coefficient, and the *p*-values were obtained from two-sided *t* test, which follows a *t*-distribution to account for smaller sample sizes. **(E)** Gene set enrichment analysis (GSEA) performed on single cells mapped to both the medullary and cortical regions using Cell2Spatial. **(F)** Left: Epithelial cell transcriptomes from a mouse kidney single-cell atlas were mapped onto spatial spots of a normal mouse kidney using Cell2Spatial, displayed with jitter within their assigned spots. Right: The same representation, with cells colored based on their known distance to the inner medulla. **(G)** Concordance between predicted and known distances of each epithelial state to the base of the inner medulla (illustrated in the left-top panel). "*R*" stands for the Spearman correlation coefficient, and the *p*-values were obtained from two-sided *t*-tests. **(H) (i)** Anatomical structures within the dorsolateral prefrontal cortex (DLPFC), **(ii)** Spatial architecture determined using Cell2Spatial, **(iii)** CytoSPACE, and **(iv)** Heat map showing the distribution of cell types in anatomical structures of the DLPFC. The intensity of the color indicates the cellular fraction within the specific anatomical region. (top) Cell2Spatial; (bottom) CytoSPACE. **(I)** Bar plots showing score-guided mapping accuracy (SGMA) metrics of various mapping tools across multiple spatial tissue samples, including human thymus, mouse kidney, DLPFC, human lung, intestinal, and breast tissues. The data underlying this figure can be found at https://doi.org/10.5281/zenodo.17212677.

at the existence of certain cTEC subtypes exhibiting characteristics of mTECs [31,34]. These suggest that the thymus architecture restored by Cell2Spatial aligned with the developmental pathway of T-cells. While CytoSPACE, CellTrek, and Tangram also partially captured the relationship between critical T-cell subpopulations and thymus structures, their performance slightly lags behind Cell2Spatial (Fig 4C). Notably, Seurat's performance was sub-optimal due to its interpretation of spots at the SC level. These findings highlight the effectiveness and reliable of Cell2Spatial in deciphering the essential spatial structures of anatomical tissues.

The correlation between the average spatial distances of different T-cell subtypes and their developmental pseudo-time were also analyzed. As shown in Fig 4D, the results of Cell2Spatial were highly consistent ($R = 0.81$), with later stages of development situated closer to the medullary center (Fig 4D and S5 Table). Additionally, we carried out a gene set enrichment analysis (GSEA) to compare single cells that mapped to the cortical and medullary regions. As shown in Fig 4E, cells mapped by Cell2Spatial to the medullary region exhibited significant enrichment in processes related to T-cell activation and adaptive immunity. Conversely, cells mapped to the cortical region displayed a strong association with cell proliferation and positive selection (Fig 4E). Thus, Cell2Spatial exhibited good ability in restoring the developmental trajectories of different cell types within a spatial context compared to similar methods.

Next, we examined whether Cell2Spatial could accurately replicate known spatial organization patterns in the mouse kidney [35,36], using both full-spectrum SC data and ST data. In the mouse kidney ST data, 1,835 spots were surveyed, with a median of 34,439 UMIs per spot and a maximum value of 74,120 UMIs. Notably, 82% of genes within these spots exhibited zero-expression values. Furthermore, the corresponding SC data (12,987 cells) encompassed 32 distinct epithelial differentiation states. As the results shown, Cell2Spatial not only reconstructed known regional structures, but also arranged almost 30 epithelial states in the established locations ($R = 0.83$) of the renal unit epithelium and collecting duct system (Figs 4F, 4G, and S4A–S4H and S1 and S6 Tables). The results were consistent with the actual anatomical structures of mouse kidney, and demonstrated Cell2Spatial's better performance compared to outcomes obtained with other tools. Finally, we scrutinized the ability of Cell2Spatial to restore the spatial architecture of the dorsolateral prefrontal cortex (DLPFC) [37] (S5A and S5B Fig and S1 Table). In the DLPFC ST dataset, 3,639 spots are present, with a median of 4,120 UMIs per spot and a maximum of 17,436 UMIs. Notably, 93.42% of genes exhibit no expression, indicating high sparsity. Additionally, the corresponding SC dataset (snRNA-seq), as reported by Mathy and colleagues [38], consists of 33,914 cells representing 17 distinct cell types. Cell2Spatial effectively mapped the major cell types to spatial spots,

aligning with previous studies [37,38], and surpassing other tools in accurately reproducing the distribution of critical cell types within the prefrontal cortex (Figs 4H and S5C–S5F).

However, it is crucial to acknowledge the absence of precise cell localization in these real ST datasets, making it difficult to quantify the performance of different tools regarding prior anatomical structures, development, and functional inferences. To address this limitation, we proposed a strategy based on hypothesis testing and Jaccard Index framework to provide a unified assessment of various performance dimensions: SGMA (Fig 3F and 3G; see "Materials and methods"). According to the SGMA metrics calculated by our method, Cell2Spatial consistently outperformed other methods across multiple datasets, including thymus, mouse kidney, DLPFC, human lung [39,40], intestine [9], and BRAC [8] (Figs 4I, S6A–S6E, and S7A–S7I and S1 and S7 Tables). In summary, based on a comprehensive assessment of seven ST datasets spanning six tissue types, Cell2Spatial stands out as a powerful tool for reconstructing cellular developmental pathways, understanding biological functions, and decoding structural localization compared to similar tools.

## Comparative evaluation of Cell2Spatial across multiple high-resolution spatial platforms

Given the limited resolution of spots containing 1–10 cells in the 10× Visium platform, a comparative evaluation of Cell2Spatial's performance in high-resolution spatial techniques was conducted using the Xenium In Situ platform, capable of mapping hundreds of transcripts at subcellular resolution [41]. For details, mouse brain data from Xenium platform was acquired, comprising 496 genes and 36,553 spots, with median UMI counts per location averaging around 207 and a minimum of 133. The Allen database's Mouse brain SC dataset, consisting of 23 cell types and 14,249 cells, was spatially mapped (S1 Table). Our analysis revealed that Cell2Spatial closely matched spatial-based RCTD [12] annotations (RCTD commonly used for accurate annotations of high-resolution ST data), accurately depicting critical brain regions such as VLMC, L2/L3 IT, L4, L5 PT, L6CT, L6b, and Oligo (Fig 5A and 5B). While CytoSPACE demonstrated good reconstruction capabilities, Tangram exhibited some inaccuracies, particularly in capturing the distribution of Oligo cells in the mouse brain (Fig 5A and 5B). Notably, CellTrek failed to handle high-resolution Xenium data. Additionally, Seurat's labeling strategy, which transforms cell type labels onto spatial spots rather than assigning individuals cells to spatial positions, was also excluded from our comparative analysis. Quantitative comparisons of Cell2Spatial, CytoSPACE, and Tangram demonstrated that Cell2Spatial achieved higher accuracy in mapping each cell type to spatial positions (Fig 5C). Although it effectively reconstructed high-resolution spatial structures, further validation is needed to confirm the accuracy of its spatial gene expression patterns. By investigated mouse brain data from Xenium, we selected the top 10 representative genes for each cell type using the *FindAllMarkers* function from Seurat and calculated their signature scores in spots (Fig 5D). The results indicated that Cell2Spatial's reconstructed gene expression patterns closely matched the real data ($R > 0.5$), surpassing the performance of CytoSPACE and Tangram (Fig 5E). To further evaluate the end-to-end performance of Cell2Spatial on high-resolution data, we used mouse brain Visium HD data comprising 492,460 spots at 8 μm resolution, representing SC-scale granularity (S1 Table). For computational efficiency and memory optimization, the dataset was down-sampled to 50,000 spots. A synthetic spatial dataset (ST) was constructed from the expression matrix and spatial coordinates of these spots, while a corresponding SC dataset was generated from the same expression matrix with cluster identities assigned through Seurat-based clustering. The SC dataset was mapped back to the synthetic ST dataset to assess mapping accuracy, defined as the proportion of cells in each cluster correctly positioned at their original spatial locations. The results revealed that Cell2Spatial effectively reconstructed the spatial distribution of cells, closely matching the ground truth (Fig 5F). Comparative analysis with CytoSPACE and Tangram demonstrated that Cell2Spatial achieved higher mapping accuracy on the Visium HD platform (Fig 5G; Average accuracies: Cell2Spatial (0.49), CytoSPACE (0.30), Tangram (0.19)) under end-to-end mapping conditions.

Slide-seqV2 achieves close to SC resolution, albeit with lower transcript capture rates. Utilizing an existing mouse hippocampus SC RNA-seq dataset generated by Saunders and colleagues [42], encompassing 16 cell types with 52,846 cells, we reconstructed the spatial architecture of the Slide-seq V2 mouse hippocampus dataset, which consist of 53,173

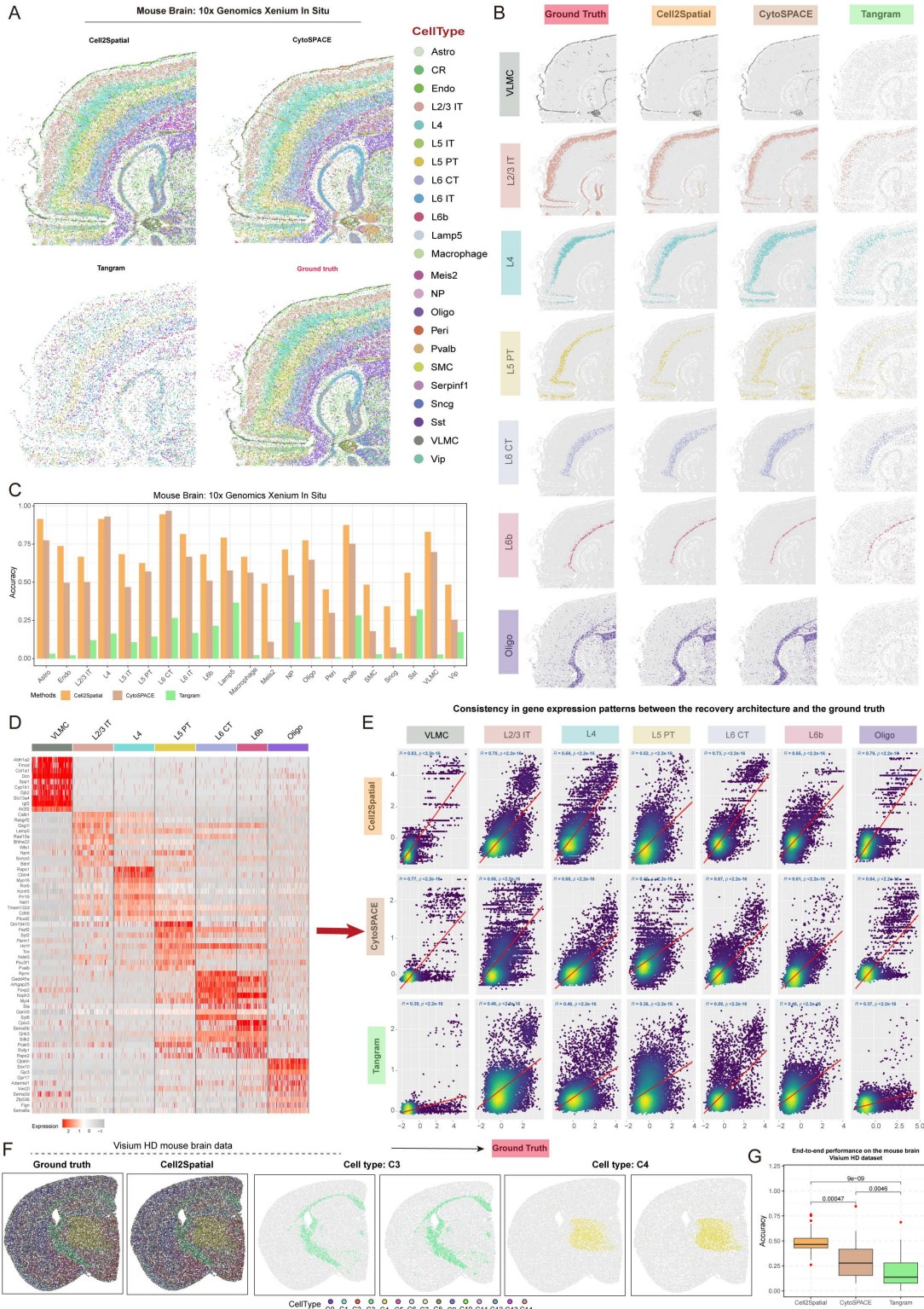

**Fig 5. Application of Cell2Spatial to 10× Genomics Xenium mouse brain data. (A)** Spatial architecture of the mouse brain reconstructed using Cell-2Spatial, CytoSPACE, and Tangram. Ground Truth (derived by RCTD [12] tool) depicts the distribution of different cell types in the reference space. Each point represents an individual cell, with distinct colors indicating various cell types. **(B)** Distributions of VLMC, L2/3 IT, L4, L5 PT, L6 CT, L6b, and Oligo

cells from the outer boundary to the inner region of the mouse brain for Ground Truth, Cell2Spatial, CytoSPACE, and Tangram, respectively. Each point denotes a single cell. **(C)** Grouped bar plot illustrating the accuracy of different cell type distributions in the Mouse Brain reconstructed by Cell2Spatial, CytoSPACE, and Tangram. Accuracy was calculated as the number of correctly mapped cells for a specific cell type divided by the total number of that cell type in the Ground Truth. **(D)** Heatmaps showing the expression distribution of the top 10 over-expressed genes for VLMC, L2/L3 IT, L4, L5 PT, L6 CT, L6b, and Oligo cell types in the 10× Genomics Xenium spatial data. Colors in the heatmap represent the expression levels. **(E)** Scatter plots demonstrate the consistency of signature scores for different cell types between the spatial architectures reconstructed by Cell2Spatial, CytoSPACE, and Tangram, compared to the Ground Truth. Each point represents a single spot. $R$ denotes the Pearson correlation coefficient, and $p$-values were derived from two-sided $t$-tests. **(F)** Evaluation of Cell2Spatial end-to-end performance using Visium HD spatial data (S1 Table) from the mouse brain. The left panel shows the true spatial distribution of cells, while the right panel presents the reconstructed distribution by Cell2Spatial. Panels C3 and C4 illustrate the spatial distribution of two selected single-cell clusters. Each point represents an individual cell. **(G)** Boxplot showing the accuracy of various mapping tools in reconstructing spatial organization using mouse brain Visium HD data. Accuracy is measured as the proportion of cells within each single-cell cluster accurately mapped to their original spatial positions. $P$-values were derived from two-sided $t$-tests. The data underlying this figure can be found at https://doi.org/10.5281/zenodo.17212677.

spots and 23,264 genes (S1 Table). Within this dataset, where 98.19% of gene expression matrix entries are 0 and the median UMI counts per location hover around 302, with a minimum of only 10, Cell2Spatial has effectively reconstructed the spatial organization of the mouse hippocampus, accurately delineating regions like the entorhinal cortex, CA principal cells, dentate principal cells, oligodendrocytes, astrocytes, and ependymal areas, aligning with spatial-based RCTD annotations (i.e., ground truth) (Fig 6A and 6B). While CytoSPACE also displayed promising reconstruction abilities, it notably misallocated numerous entorhinal cortex cells to the CA principal cells region. Conversely, Tangram exhibited a disorderly distribution pattern (Fig 6A and 6B). Further quantitative analysis confirmed Cell2Spatial performed better overall compared to CytoSPACE and Tangram (Fig 6C). Additionally, in another Slide-seq V2 dataset comprising 11,626 positions of mouse cerebellum (S1 Table), with 98.55% of gene expression matrix entries being zero and median UMI counts per location approximately 30,014, Cell2Spatial accurately assigned cell type labels and captured the multi-layered structure of the cerebellum, such as Bergmann, Granule, and Purkinje, consistent with RCTD annotations (Fig 6D–6F). The analysis of cellular compositions in reconstructed spatial structures demonstrated Cell2Spatial's high consistency with RCTD-derived cellular compositions ($R > 0.97$), while CytoSPACE and Tangram lagged significantly behind, underscoring their inadequacy for low-capture-rate spatial data (Fig 6G and 6H).

Overall, these findings strongly indicate that Cell2Spatial remains robust in its performance even when processing high-resolution spatial data, even with lower transcription capture rates.

## Diverse applications of Cell2Spatial

To investigate the potential applications of Cell2Spatial, we utilized it to analyze ST data (comprising 1,438 spots) of mouse kidney tissue, aiming to explore the spatial co-existences among various cell types and the temporal differentiation of crucial cell types within the spatial framework. By assigning single cells from the mouse kidney (12,987 cells encompassing 13 cell types) reported by Ransick and colleagues [36] to spots, as illustrated in Fig 7A, distinct spatial distributions among different cell types were observed (Figs 7A, S8A, and S8B and S1 Table). To quantify these relationships, we introduced a co-existence index, and the results revealed robust spatial co-localization links between T cells, fibroblasts, and natural killer (NK) cells (Fig 7B and S8 Table; see "Materials and methods"). To further verify these findings, we highlighted the expression of their respective marker genes (*Cd3d*, *Col3a1,* and *Nkg7*) in the ST data of mouse kidney, which showed a significant overlap in the spatial distribution of these three cell types, confirming the results of Cell2Spatial (S8C Fig). Additionally, we elucidated the differentiation timelines of proximal tubular and distal tubular cells through pseudo-time analysis using Monocle2 [43–45]. As shown in Fig 7C and 7D, proximal tubules exhibited an outward-to-inward differentiation trend, while distal tubules displayed an inward-to-outward progression (Fig 7C and 7D). The opposing timelines align with the findings of Wei and colleagues [3]. Thus, Cell2Spatial can accurately track the timelines of cell differentiation.

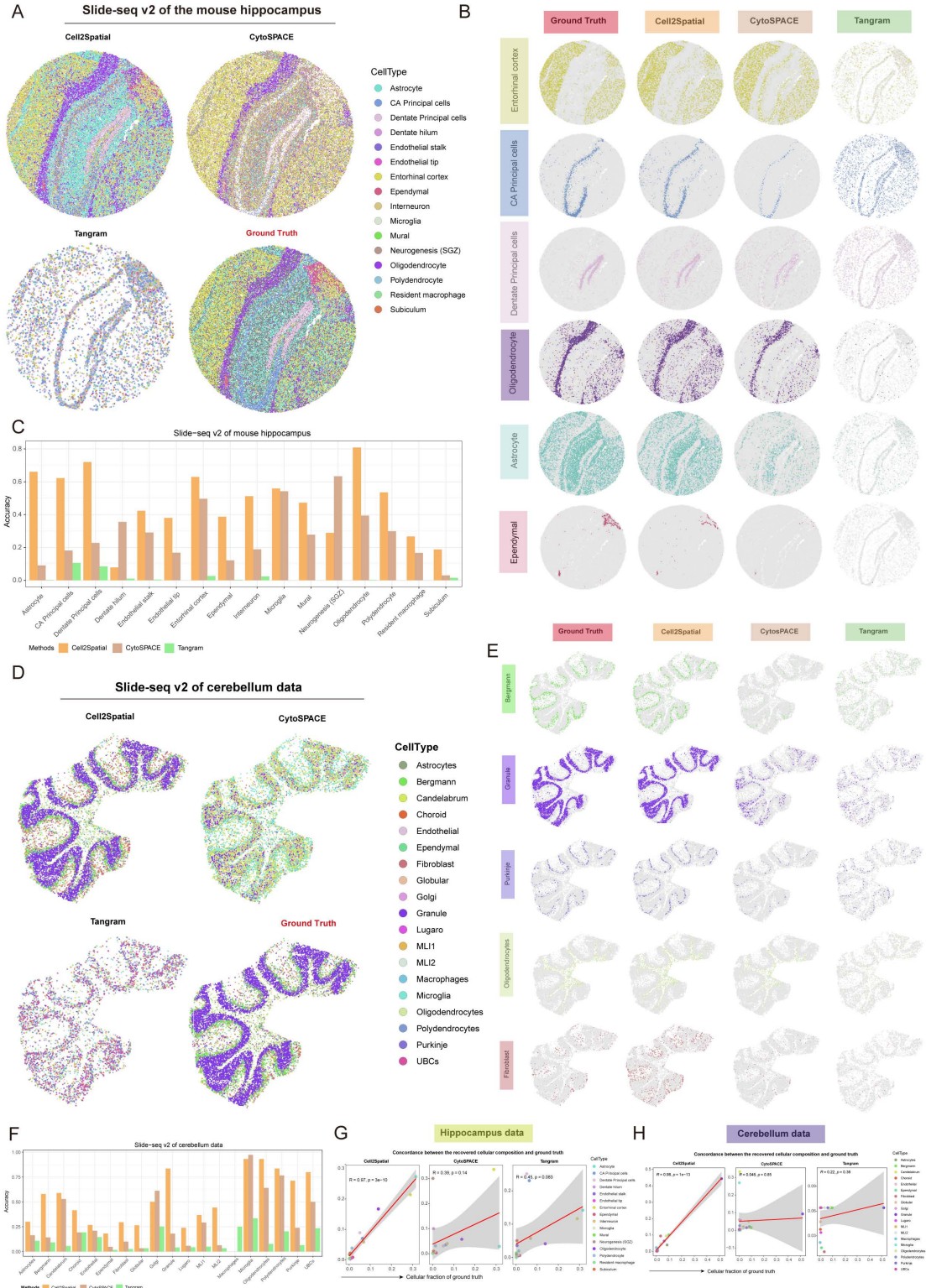

**Fig 6. Application of Cell2Spatial to Slide-seq V2 hippocampus and cerebellum data.** **(A)** Spatial architecture of the mouse hippocampus reconstructed using Cell2Spatial, CytoSPACE, and Tangram. Ground Truth (derived by RCTD [12] tool) represents the distribution of different cell types in the reference space. Each point represents a single cell, with different colors indicating different cell types. **(B)** Distributions of Entorhinal cortex, CA

Principal cells, Dentate Principal cells, Oligodendrocytes, Astrocytes, and Ependymal cells for Ground Truth, Cell2Spatial, CytoSPACE, and Tangram. Each point represents a single cell. **(C)** Grouped bar plot showing the accuracy of cell type distributions in the hippocampus reconstructed by Cell2Spatial, CytoSPACE, and Tangram. Accuracy was calculated as the number of correctly mapped cells for a specific cell type divided by the total number of that cell type in the Ground Truth. **(D)** Spatial architectures of the cerebellum reconstructed using Cell2Spatial, CytoSPACE, and Tangram. Ground Truth represents the distribution of different cell types in the reference space. Each point represents a single cell, with distinct colors indicating different cell types. **(E)** Distributions of Bergmann cells, Granule cells, Purkinje cells, Oligodendrocytes, Astrocytes, and Fibroblasts for Ground Truth, Cell2Spatial, CytoSPACE, and Tangram. Each point represents a single cell. **(F)** Grouped bar plot showing the accuracy of different cell type distributions in the cerebellum reconstructed by Cell2Spatial, CytoSPACE, and Tangram. Accuracy was calculated as the number of correctly mapped cells for a specific cell type divided by the total number of that cell type in Ground Truth. **(G, H)** Scatter plots showing the consistency between the spatial cellular compositions recovered by Cell2Spatial, CytoSPACE, and Tangram, compared to Ground Truth for both the hippocampus (G) and cerebellum (H). Each point represents a cell type. *R* indicating the Pearson correlation coefficient (PCC), with *p*-values obtained from two-sided *t*-tests, following a *t*-distribution to account for smaller sample sizes. The data underlying this figure can be found at https://doi.org/10.5281/zenodo.17212677.

To determine whether Cell2Spatial can detect critical structural regions within tissues, ST data with 3,206 spots concerning tertiary lymphoid structures (TLS) associated with renal cell cancer (RCC) from the study published by Meylan and colleagues [46] were acquired (Fig 7E and S1 Table). TLSs are closely associated with tumor development and metastasis, carrying diagnostic and therapeutic significance in cancer. In this dataset, the precise coordinates of TLS are well-documented, with a notable concentration in Cluster 3 and Cluster 4 (Fig 7E and 7F). By superimposing an immune SC atlas (down-sampled to 24,834 cells covering 25 major immune cell types) [47] onto the ST section of RCC, a diverse array of immune cell types within the TLS were observed (Fig 7G). Spatial cell co-existence analysis revealed that B cells were the major interacting cell type (Fig 7H and S9 Table), and had the most substantial and abundant interactions with other immune cells in Cluster 4, closely followed by Cluster 3 (Fig 7I and 7J). These findings suggest the presence of a functionally significant structure (Clusters 3 and 4) in RCC tissue, in alignment with previously documented structures. Taken together, Cell2Spatial can used to explore functional and structural regions within tissues, and the intricate network of interactions between diverse cell types and their spatial arrangement.

## Discussion

This study underscores the utility of Cell2Spatial as a powerful tool for mapping individual cells from SC datasets to spatial spots within ST data, simplifying the generation of spatial cellular maps. In contrast to traditional ST deconvolution methods [10,12,15], Cell2Spatial enables a detailed investigation of each cell's specific characteristics in relation to spatial spots. While Cell2Spatial shares similarities with established spatial mapping methods like CytoSPACE [20], CellTrek [3], Tangram [19], and Seurat [18], we emphasize its unique attributes. Cell2Spatial utilizes a spatially weighted maximum likelihood model to assess the similarity between cells and spatial spots, ensuring accurate representation of spatial relationships within tissues and enhancing the detection of cellular heterogeneity. When cell types in SC and ST datasets are not fully congruent, Cell2Spatial employs a hotspot detection strategy to detect the spatial distribution of specific cell types. This approach facilitates the exclusion of single cells corresponding to absent cell types in the ST dataset, thereby improving the reliability of spatial mapping. By selecting "cell-count-regulated genes," Cell2Spatial accurately estimates cell numbers in each spot, enabling precise reconstruction of cell type composition within spatial contexts. Additionally, Cell2Spatial employs a FNN to preassign cells to spatial clusters based on a joint embedding of SC and ST data. This methodology optimizes computational efficiency and memory usage, significantly enhancing overall performance. Cell2Spatial thus effectively reconstructs the spatial architecture of tissue slices from SC data, and addresses the complexities of spatial mapping between SC and ST datasets when cell types are not perfectly aligned.

According to performance evaluations that encompassing a range of tissue ST data types from the 10× Visium platform with relatively low resolution in spot, Cell2Spatial demonstrated good performance in accurately recovering spatial architectures, gene expression, and cellular compositions, outperforming the capabilities of existing tools. Notably, for

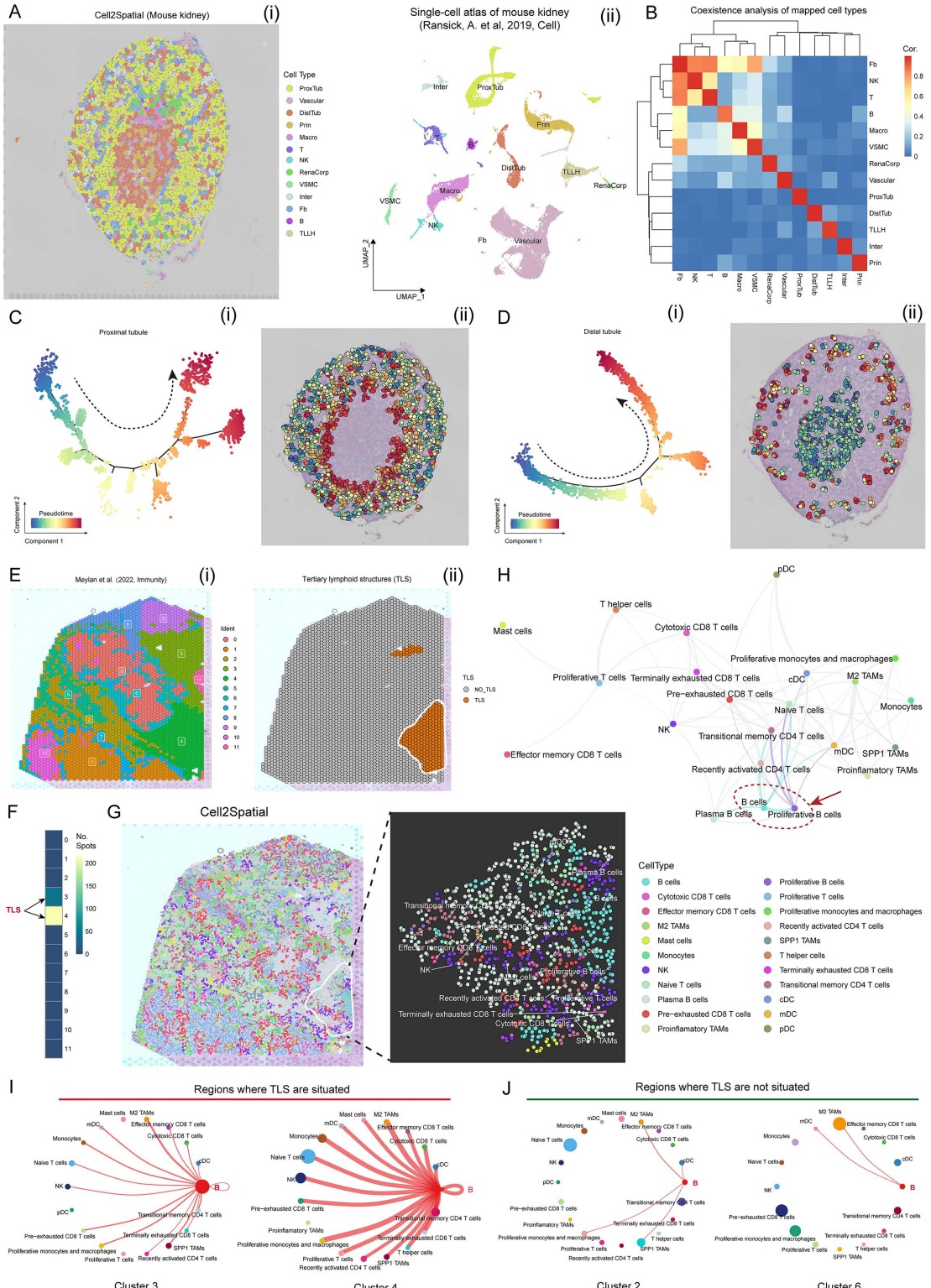

**Fig 7. Cell2Spatial enables versatile applications across various analytical domains.** (A) Left: Spatial architecture of the mouse kidney depicted by Cell2Spatial. Each dot represents an individual cell, with cell types are color-coded. Right: Uniform Manifold Approximation and Projection (UMAP) plot showing the single-cell atlas of mouse kidney. **(B)** Heat map showing the coexistence of mapped cell types in spatial spots. The Pearson correlation

coefficient (PCC) is obtained based on coexistence index (see "Materials and methods"). **(C)** Trajectory analysis of proximal tubule cells (left), with spatial mapping of the pseudo-time values in the tissue section (right). **(D)** Trajectory analysis of distal tubule cells (left), with spatial mapping of the pseudo-time values in the tissue section (right). **(E)** Left: Renal cell cancer (RCC) tissue section displaying spot clustering determined using the Seurat common processing strategy. Right: Pre-defined spatial coordinates of the tertiary lymphoid structure (TLS) for reference. **(F)** Heat map showing the relationship of inclusion between TLS and spatial clusters. The intensity of color indicates the number of TLS spots located in the cluster. **(G)** Left: Spatial architecture of the RCC tissue section reconstructed using Cell2Spatial. Each dot represents an individual cell and cell types were color-coded. Right: Enlarged cell atlas located within the TLS region. **(H)** Overview of cell types co-existence in the RCC tissue section. Edge thickness reflects the strength of the co-existence index, with edges below a specified cutoff (0.05) excluded. **(I)** Circle plots showing the interactions between B cells and other cell types within an independent spot cluster. The thickness of the edges indicates the strength of interactions. Left: Cluster 3; Right: Cluster 4. **(J)** Circle plots depicting the interactions between B cells and other cell types within an independent spot cluster. Left: Cluster 2; Right: Cluster 6. The data underlying this figure can be found at https://doi.org/10.5281/zenodo.17212677.

thymus spatial data, Cell2Spatial effectively reconstructed developmental pathways, elucidated biological functions, and provided insights into structural positioning. Moreover, when utilizing high-resolution ST data (approaching SC or subcellular levels), including 10× Xenium and Slide-seq V2, Cell2Spatial exhibited outstanding capabilities in reconstructing spatial architectural structures. Even in ST data with lower transcript capture rates, Cell2Spatial maintains robustness and accuracy, outperforming CytoSPACE and Tangram. Particularly, its performance in analyzing mouse kidney tissue emphasizing its strong capacity to track the temporal evolution of key cell types. By overlaying an immune SC atlas with RCC ST data, Cell2Spatial unveiled intricate interactions among distinct cell types and identified disease-related regions, i.e., TLS.

Overall, through reconstructing tissue spatial architectures at SC resolution, Cell2Spatial provides enhanced biological insights, such as differential expression analysis, developmental trajectories, cell–cell interactions, and complex spatial gene expression patterns.

Although our study primarily concentrated on the utilization of SC data to investigate potential spatial architectural structures within 10× Visium, Xenium, HD Visium, and Slide-seq V2 ST data, Cell2Spatial is a reference-based method that can be applied to diverse spatial data from various sources, such as MERSCOPE [48], SMART-Seq2 [49], and Drop-Seq [50]. It's essential to note that while using Cell2Spatial, its performance may be influenced by several underlying factors. Firstly, Cell2Spatial relies on specificity entropy to identify representative gene sets specific to cell types, which requires well-annotated or clustered SC data for accurate identification. Secondly, measuring similarity between individual cells and spatial spots may introduce biases, especially for cell types with similar evolutionary lineages, leading to discrepancies between their mapped spatial positions and true distributions. Moreover, RNA spread in ST sequencing may cause uncorrected quantitative biases in the ST data. Specifically, Cell2Spatial shows diminished performance in end-to-end mapping, likely due to insufficient cell type-specific markers to effectively differentiate variations among cells of the same type. Additionally, the limitations of the Linear Assignment Algorithm in capturing non-linear relationships between single cells and spatial spots contribute to this performance drop. For sub-cellular resolution datasets, integrating high-resolution tissue imaging with DAPI staining for accurate segmentation could improve mapping precision. Moreover, developing algorithms that exploit sub-cellular resolution transcriptomic signals would enable a more detailed exploration of spatial heterogeneity within individual cells, further enhancing Cell2Spatial's capabilities. These limitations highlight an area for improvement in future research.

Despite these challenges, Cell2Spatial has the potential to become a valuable tool for biological research, advancing the study of cellular and tissue spatial structures in the context of human diseases.

## Materials and methods

### 1. Cell2Spatial analytical framework

**Data collection and normalization.** The SC reference expression profile is assumed to be an $N \times C$ matrix, with $N$ representing genes and $C$ representing cells. The dataset is well-annotated (or clustered) into $k$ distinct cell types (or

clusters). Any cell type with fewer than a predefined threshold of cells (default: 5 cells) was excluded from the SC dataset. The spatial transcriptome (ST) expression profile is an $M \times S$ matrix, where $M$ represents genes and $S$ represents spots. A total of $G$ genes is shared between the SC and ST. Data normalization was achieved by applying Seurat's *SCTransform* method to both the SC ($X \to G \times C$) and ST data ($Q \to G \times S$).

**Cell-type-specific genes.** Given that most genes cannot effectively distinguish among cell types, specific genes corresponding to distinct cell types were identified with computational efficiency using a specificity metric inspired by Shannon entropy structure (S9A Fig). Initially, the SC profile $X$ was used to generate the average expression matrix $X^*$ ($X^* \to G \times K$), with rows representing genes and columns representing cell types. Each entry in $X^*_{ij}$ represents the mean expressions level of gene $i$ in cell type $j$. Gene specificity scores were then computed using the following formula:

$$spec_i = \frac{w_i}{K} \cdot \sum_{j=1}^{K} \left( \frac{X^*_{ij}}{\overline{X^*_{i.}}} \cdot \log_2 \frac{X^*_{ij}}{\overline{X^*_{i.}}} \right)$$

(1)

Here, $spec_i$ represents the value characterizing the cell-type specificity of gene $i$ (higher values indicate greater gene specificity), $\overline{X^*_{i.}}$ represents the mean expression of gene $i$ across all cell types. The summation yields zero if the gene $i$ is uniformly expressed across cell types, and reaches a maximum value of $K\log_2 K$ when it is exclusively expressed in a single cell type. The weight $w_i$ for gene $i$, was calculated as:

$$w_i = \frac{\max\limits_{j \in \{1,...,K\}} X^*_{ij} - \min\limits_{i' \in 1,...,G} \left( \max\limits_{j \in \{1,...,K\}} X^*_{i'j} \right)}{\max(X^*) - \min\limits_{i' \in 1,...,G} \left( \max\limits_{j \in \{1,...,K\}} X^*_{i'j} \right)}$$

(2)

Thus, the gene weights $w$ are directly proportional to the expression level and serve as a key factor in determining the specificity values. Here, $\max\limits_{j \in \{1,...,K\}} X^*_{i'j}$ denotes the maximum average expression of gene $i$ across all cell types, $\min\limits_{i' \in 1,...,G} \left( \max\limits_{j \in \{1,...,K\}} X^*_{i'j} \right)$ represents the smallest of these maximum average expression levels across all genes, and $\max(X^*)$ is overall maximum element in the matrix $X^*$, encompassing all genes and all cell types. Each gene was initially assigned to the cell type in which it showed the highest average expression. For each cell type, we then calculated the expression variance of all genes across individual cells. Among the genes assigned to that cell type, those with variances exceeding the mean variance (calculated across all genes) by more than three standard deviations were excluded, to eliminate genes with unusually high within-type variability.

For each cell type, the remaining assigned genes that passed the variance filter were ranked based on their specificity scores, and the top 100 genes were selected by default. This resulted in a gene set $\mathcal{Z} = \{Z_j\}_{j=1}^{K}$, where $Z_j$ represents the set of specific genes for cell type $j$. The total number of selected genes, denoted as $m$ ($m \leq 100 \times K$, as some cell types may contain fewer than 100 qualified genes after filtering).

**Measuring similarity between single cells and spots.** To gauge the similarity between single cells and spatial spots, the specific gene list $\mathcal{Z}$ was initially converted into a vector and duplicates were removed, resulting in a length of $m$ as previously defined. Then, the probability of specific gene $i$ for each cell $j^*$(i.e., $p_{ij*}$), was estimated using $X_{ij*}$ after applying Laplace Smoothing (by adding 1e-6 to account for zero values). The formula used was: $p_{ij*} = \frac{10^{-6} + X_{ij*}}{\sum_i (10^{-6} + X_{ij*})}$, ensuring that $\sum_i p_{ij*} = 1$. Based on the multinomial assumption, the likelihood function for a spot $t$ with profile $(y_1, \ldots, y_i, \ldots, y_m)$ to cell $j^*$ was calculated as $\frac{(\sum_i y_i)!}{\prod_i (y_i!)} \prod_i \left( p_{ij*}^{y_i} \right)$, and can be further described as follows:

$$p(t|j^*) = \frac{(\sum_i y_i)!}{\prod_i (y_i!)} \prod_i \left( p_{ij*}^{y_i} \right) \propto \prod_i \left( p_{ij*}^{y_i} \right)$$

(3)

Given that the geometric spatial distance can provide varying viewpoints to reflect the association between single cells and spots, Seurat's CCA strategy was used for the integration of SC and ST datasets. Subsequently, UMAP was applied to obtain a two-dimensional projection of the integrated data. Although UMAP's embeddings may distort global distances to enhance visualization, its local structure preservation ensures a meaningful representation of the relationships between a spatial spot and its neighboring cells, helping to characterize tissue architecture and cellular distributions [51] (S9B Fig; performance assessment of distance weighting versus no weighting for Cell2Spatial). Thus, the Euclidean distance between each spot and single cell was calculated, resulting in a distance matrix $\boldsymbol{D}_{S \times C}$ composed of $S$ spots and $C$ cells. A scaled distance-based weight matrix was then derived using the following formula:

$$\boldsymbol{D}'_{tj*} = \frac{quantile\left(\boldsymbol{D}_{t\cdot}, probs\right) - \boldsymbol{D}_{tj*}}{quantile\left(\boldsymbol{D}_{t\cdot}, probs\right) - \min(\boldsymbol{D}_{t\cdot})} \tag{4}$$

where $\boldsymbol{D}_{t\cdot}$ is a vector representing the distances between spot $t$ and all single cells in UMAP coordinates, $quantile\left(\boldsymbol{D}_{t\cdot}, probs\right)$ denotes the quantile value of the distance vector $\boldsymbol{D}_{t\cdot}$ at the specifies probability level (with $probs$ constrained between 0.01 and 1, defaulting to 0.3) (S9C Fig). Notably, if spot $t$ is closer to cell $j*$ in the UMAP projection, the weight $\boldsymbol{D}'_{tj*}$ is higher. If $\boldsymbol{D}'_{tj*}$ is below 0, it is re-set to 0. To combine these distance weights further, the likelihood similarity incorporating spatial distance-based weights was calculated as follows:

$$\boldsymbol{P}' = \left( \frac{\log_{10} \boldsymbol{P}_{tj*} - \min(\log_{10} \boldsymbol{P}_{t\cdot})}{\max\left(\log_{10} \boldsymbol{P}_{t\cdot}\right) - \min(\log_{10} \boldsymbol{P}_{t\cdot})} \right)_{t=1,\dots,S;\ j*=1,\dots,C} \times \boldsymbol{D}' \tag{5}$$

Here, $\boldsymbol{P} = [p\,(t|j*)] \in \mathbb{R}^{S \times C}$ is the original likelihood matrix representing the similarity between spots and cells, and $\boldsymbol{P}' = [p'\,(t|j*)] \in \mathbb{R}^{S \times C}$ is the final similarity matrix after incorporating spatial information. $\boldsymbol{P}_{t\cdot}$ is the vector of observed likelihoods for spot $t$ across all cells. $\boldsymbol{D}'$ is the corresponding normalized spatial distance weight matrix of the same dimensions, and "$\times$" indicates element-wise multiplication.

**Determination of spatial hotspot regions of cell types.** To detect spatial hotspots for each cell type, the Getis-Ord $G*$ index was employed, involving several steps: Initially, spatial coordinates of spots within the tissue sample were extracted from the ST Seurat object. Then, a neighborhood structure was established using k-nearest neighbors, where each spot's nearest neighbors (default: 5 spots) were determined to form its neighborhood set. Next, the *AddModuleScore* function from Seurat was used to calculate signature scores for spots based on the gene sets specific to that cell type in $\mathcal{Z}$. Finally, the statistic $G*$ was calculated for each spot using the following formula:

$$\boldsymbol{G}^*_{tj} = \frac{\sum_{t'=1}^{S} \theta_{tt'}\left(\boldsymbol{x}_{t'} - \overline{\boldsymbol{x}}\right)}{\Delta \sqrt{\frac{\left(S \sum_{t'=1}^{S} \theta_{tt'}^2\right) - \left(\sum_{t'=1}^{S} \theta_{tt'}\right)^2}{(S-1)}}} \tag{6}$$

where $\boldsymbol{G}^*_{tj}$ is the $G*$ index of spot $t$ in cell type $j$, $\boldsymbol{x}$ represents the vector of signature scores for cell type $j$ across all the spots, $\boldsymbol{x}_{t'}$ denotes the score for spot $t'$, $\overline{\boldsymbol{x}}$ is the global mean score of the ST section. $\theta_{tt'}$ is a binary weight indicating whether spot $t'$ is a neighbor of spot $t$ (1 if true, 0 otherwise), $\Delta$ is the standard deviation of the signature scores in $\boldsymbol{x}$, and $S$ represents the total number of spots as previously defined. $\sum_{t'=1}^{S} \theta_{tt'}\left(\boldsymbol{x}_{t'} - \overline{\boldsymbol{x}}\right)$ represents the difference between the signature scores of neighboring spots and the global mean $\overline{\boldsymbol{x}}$. Denominator $\Delta \sqrt{\frac{\left(S \sum_{t'=1}^{S} \theta_{tt'}^2\right) - \left(\sum_{t'=1}^{S} \theta_{tt'}\right)^2}{(S-1)}}$ normalizes the numerator by adjusting for the scale of the weighted differences. The standard $\boldsymbol{G}*$ is essentially a *z-score* that measures how the neighborhoods of spot $t$ deviate from the context mean score [52]. Therefore, the *p*-value for each $\boldsymbol{G}^*_{tj}$ statistic of cell type $j$ was computed as:

$$p_{tj}^* = 1 - \Phi\left(\boldsymbol{G}_{tj}^*\right)$$

(7)

where $\Phi$ is the cumulative distribution function of the standard normal distribution (S9D Fig). Spot $t$ was classified as a hotspot for cell type $j$ if $p_{tj}^* \leq \alpha$ (default: $\alpha = 0.05$). The processes mentioned above were applied to all spots, resulting in a Boolean matrix $\boldsymbol{H}_{S \times K}$ with dimensions $S$ (spots) by $K$ (cell types). Each entry in this matrix indicates whether a given spot is a hotspot (1) or not (0) for a specific cell type. This yields a list of detected hotspots for each cell type, denoted as $\mathcal{H} = \{\mathcal{H}_j\}_{j=1}^{K}$, with a unique length of $S'$. $\mathcal{H}_j$ represents the set of detected hotspots belonging to cell type $j$.

**Clustering spots of ST data.** Following the standard Seurat [53] workflow for clustering ST data, the *SCTransform* function was used for normalization and variance stabilization. *SCTransform* computes Pearson residuals for each gene, reflecting deviations from expected expression based on a regularized negative binomial model. Genes with high variance in these residuals are considered highly variable. By default, the top 3,000 HVGs were used for clustering, which was performed with the resolution parameter set to 0.2 (as default). The obtained ST clusters were defined as a list of set $\mathcal{C} = \{\mathcal{C}_r\}_{r=1}^{R}$, where $\mathcal{C}_r$ represents spots belonging to the $r$-th cluster.

**Estimation of cellular compositions in spatial section.** To examine the cellular composition within a spatial section, SC expression profile was used as a reference, with $m$ cell-type-specific genes in $\mathcal{Z}$ serving as distinctive features. This process generated a base matrix $\boldsymbol{B}_{m \times K}$, where rows represent genes, columns represent cell types, and each entry represents the average expression of a specific gene in the corresponding cell type. A reference-based deconvolution strategy, inspired by the LinDeconSeq tool [21,54], was established using a RLM to predict the relative prevalence of distinct cell types within each spot. To enhance estimation accuracy across the entire ST section, the expression vector of spot $t$ was smoothed by the averaged expression of the six (as default) nearest neighbors within the same ST-defined cluster, including spot $t$ itself, resulting in an updated vector $\boldsymbol{y} = (y'_1, \ldots, y'_i, \ldots, y'_m)$. This smoothing strategy leverages the spatial coherence of gene expression and cell type composition within local regions, while mitigating estimation bias caused by uneven cell numbers across spots [55,56] (S9E Fig). Then, incorporating RLM to link $\boldsymbol{y}$ and $\boldsymbol{B}_{m \times K}$ allows for the estimation of cellular proportions $\boldsymbol{f}_t$ of spot $t$ by minimizing the squared discrepancies between observed data.

$$\min_{\boldsymbol{f}_t} \|\boldsymbol{y} - \boldsymbol{B}\boldsymbol{f}_t\|_2$$

(8)

where $\boldsymbol{f}_t$ is the non-negative cellular proportion vector to be estimated for spot $t$, with dimension $1 \times K$, and $\|\boldsymbol{f}_t\|_1 = 1$.

To estimate the overall cell type composition of the ST tissue section, cellular proportions are typically aggregated across all spatial spots. Since spots within the same spatial cluster tend to share similar cellular profiles, reference-based deconvolution methods may assign low, non-zero proportions to cell types that are genuinely absent in certain clusters. While these small residual estimates are individually minor, their cumulative effect across multiple spots can lead to systematic biases in the global cell type composition inference. To address this, a correction factor based on the hotspot matrix $\boldsymbol{H}_{S \times K}$ was introduced, calculated as follows:

$$\boldsymbol{V} = \left( rescale \left( \sum_{spot\, t \in \mathcal{C}_r} \boldsymbol{H}_{tj} \right)_{j=1}^{K} \right)_{r=1}^{R}$$

(9)

Here, $\mathcal{C}_r$ represents the $r$-th cluster to which spot $i$ belongs. The expression $\left( \sum_{spot\, t \in \mathcal{C}_r} \boldsymbol{H}_{tj} \right)_{j=1}^{K}$ is a vector of length $K$ that reflects the number of times each cell type significantly appears in the cluster $\mathcal{C}_r$. This vector then was rescaled to a range of 0–1 using the *rescale* function (i.e., Min-max normalization) from "scales" R package (version 1.3.0). Each element of $\boldsymbol{V}$ is then binarized using a predefined threshold $\tau$ (default 0.01) as follows:

$$V_{r,j} = \begin{cases} 1, & \text{if } V_{r,j} \geq \tau \\ 0, & \text{otherwise} \end{cases} \tag{10}$$

The resulting matrix $V$ is a wrapped correction factor matrix with dimensions $R \times K$ ($R$ represents the number of spot clusters as previously defined). Therefore, the cellular composition $f_t$ can be adjusted using the following formula:

$$\frac{f_t \times V_{r\cdot}}{f_t \cdot V_{r\cdot}^T} \tag{11}$$

$V_{r\cdot}$ represents the correction factors of cluster $\mathcal{C}_r$. $V_{r\cdot}^T$ denotes the transpose of the $r$-th row vector of matrix $V$, allowing for a valid dot production. The cellular proportions of the spots in $\mathcal{H}$ (excluding spots not classified as hotspot for any cell type) are organized into a matrix $F_{S' \times K}$, where each row represents the adjusted compositions of one spot. Finally, the cellular compositions for the entire ST slice were derived by following formula.

$$f^* = \frac{\left(\sum_{t=1}^{S'} F_{tj}\right)_{j=1}^{K}}{\sum_{j=1}^{K} \sum_{t=1}^{S'} F_{tj}} \tag{12}$$

$f^*$ is the final estimated cellular proportion vector of length $K$ for the entire ST section (S9F Fig).

**Estimation of cell counts per spot for low-resolution spatial data using a corrected saturation model.** To infer the number of cells per spot in low-resolution ST data, we applied a corrected saturation model that accounts for spot-level library size and gene count, assuming that the number of detected genes per spot follows a saturation curve as cell numbers increase. Specifically, the spatial expression matrix $Q'$ records UMI counts for $G$ genes across $S'$ spots, where $Q'_{it}$ represents the UMI count of gene $i$ in spot $t$. To reduce noise, genes with low expression were excluded by applying a spot-level threshold (e.g., UMI ≥ 2), retaining only those with reliable signals. For each spot $t$, the library size is defined as $L_t = \sum_{i=1}^{G} Q'_{it}$, and the gene count is calculated as $\widetilde{G}_t = \sum_{i=1}^{G} I\left(Q'_{it} > 1\right)$, where $I$ is an indicator function.

Assuming the maximum number of cells among spots is $n^{max}$ (e.g., 10 for 10× Visium), we modeled the relationship among the gene count $\widetilde{G}_t$, cell number $c_t^{gsize}$, and library size $L_t$ using a corrected saturation model: $\widetilde{G}_t = \widetilde{G}_{max}\left[1 - \left(1 - \widetilde{g}/\widetilde{G}_{max}\right)^{c_t^{gsize}}\right]$. $\widetilde{G}_{max}$ is the maximum observed gene count across all spots, and $\widetilde{g}$ represents the average number of genes per cell. To estimate $c_t^{gsize}$, we initially computed a composite score for each spot to balance library size and gene count:

$$w_L \cdot \frac{L_t}{L_{max}} + w_{\widetilde{G}} \cdot \frac{\widetilde{G}_t}{\widetilde{G}_{max}} \tag{13}$$

where $w_L$ and $w_{\widetilde{G}}$ are the weights for library size and gene count (default: 0.5). The reference spot was estimated from the mean of spots exceeding the quantile cutoff of the composite score (90% by default) and was assumed to contain $n^{max}$ cells. The corresponding reference library size $L_{ref}$ and gene count $\widetilde{G}_{ref}$ were then used to compute g: i.e., $\widetilde{g} = \frac{\widetilde{G}_{ref}}{n^{max}}$. Subsequently, the cell number for spot $t$ was obtained by solving:

$$c_t^{gsize} = \frac{\log\left(1 - \widetilde{G}_t/\widetilde{G}_{ref}\right)}{\log\left(1 - \widetilde{g}/\widetilde{G}_{ref}\right)} \tag{14}$$

To ensure consistency, we incorporated the library size constraint $c_t^{gsize} \approx c_t^{libsize} = L_t / l$, where $l = \frac{L_{ref}}{n^{max}}$. Finally, we calculated the harmonic mean of the $c_t^{gsize}$ and $c_t^{libsize}$, and rounded to the nearest integer. i.e.,

$$c_t^* = round\left( \frac{2}{\frac{1}{\max(c_t^{gsize}, \ 1)} + \frac{1}{\max(c_t^{libsize}, \ 1)}} \right).$$ $c^*$ is a vector of cell counts across all $S'$ spots. To maintain biological plausibility and adhere to platform resolution, the $c^*$ was constrained to the range $[1, n^{max}]$. Notably, for high-resolution spatial data at the SC level, set "max.cells.in.spot = 1" (or "platform.res = High") to ensure one cell per spot. For subcellular-resolution platforms, such as STOMICS and Visium HD, adjust the bin size to approximate the dimensions of a single cell (e.g., for Visium HD, set bin.size = 8 $\mu$m).

**Assigning single cells to spatial spots.** Mapping single cells to spatial spots enables the reconstruction of spatial architecture at SC resolution for low-resolution data and high gene capture expression patterns for high-resolution data. The mapping process involves the following steps:

(1) **Update SC Seurat object**: Using the previously mentioned cellular fractions $f^*$ and spot cell $c^*$, the number of cells in each cell type was calculated as $c' = \lceil f^* \times \|c^*\|_1 \rceil$. For the $j$-th cell type, $c_j'$ cells were down-sampled from the corresponding cell type in the SC data $X$. If the cell counts of the $j$-the cell type in $X$ was less than $c_j'$, sampling with replacement was applied; otherwise, sampling without replacement. Gaussian noise ($mean = 0$, $sd = 1$) was added to the gene expression of duplicated cells to avoid a singular matrix. The updated SC data denoted as $X' \rightarrow G \times C^*$, where $C^* = \|c^*\|_1$.

(2) **Neural network design for pre-assigned individual cells to ST clusters**: Recognizing the high algorithmic complexity involved in large-scale SC allocation, global cell-to-spot mapping becomes computationally intensive and less scalable. To address this, we introduced the FNN framework to preassign single cells mentioned in step (1) to ST clusters, effectively decomposing the original global mapping task into multiple subspace mapping problems. Specifically, reference-based coembedding approach from the Seurat package was used with SC data ($X' \rightarrow G \times C^*$) as the reference and the ST data ($Q' \rightarrow G \times S'$) as the query. From the coembedded data, a four-layer neural network was trained. The input layer comprised principal components (PCs)-embedded ST data (default 30 PCs), followed by a fully connected hidden layer with 64 neurons and a rectified linear unit (*ReLU*) activation function. The subsequent two hidden layers consisted of 128 units each. To address overfitting and enhance computational efficiency, dropout (*rate* = 0.2), batch normalization, and *L2* regularization techniques were applied at each hidden layer. The output layer was based on the *softmax* function and the number of neurons in the aligned with ST clusters. The FNN's learning rate was set to 0.000001. The involved major processes as follows:

$$\left(A^{*(v)}\right)_{v=1}^3, \ \hat{Y} = \left(ReLU\left(W^{(v)} \cdot A^{(v-1)} + b^{(v)}\right)\right)_{v=1}^3, \ Softmax(W^{(4)} \cdot A^{(3)} + b^{(4)}) \tag{15}$$

$A^{(0)}$ represents the input layer of PCs of ST data. Notably, at the start of training, cells were randomly divided into a training set (95%) and a test set (5%). The test set was utilized at each epoch to evaluate the model's performance and generalization capability. The training procedure adopted a 5-fold cross-validation approach, aiming to minimized the cross-entropy loss, which evaluates the similarity between predicted and actual categories.

$$\sum_{t=1}^{S'} \sum_{r=1}^{R} I(spot_t \in \mathcal{C}_r) \log\left(\widehat{p_{tr}}\right) + \lambda\left(\sum_{v=1}^{4} \left\|W^{(v)}\right\|_2^2\right) \tag{16}$$

$I(spot_t \in \mathcal{C}_r)$ is a binary indicator if ST cluster $r$ is correct label for spot $t$, $\hat{p}_{tr}$ is the predicted probability of observation spot $t$ for cluster $r$. $\lambda$ is the regularization parameter controlling the impact of the regularization term on the loss function, with

a default value of 0.01. After completing the training of the FNN model, the co-embedded SC data was input into the model to determine the probability of each cell belonging to specific ST cluster. The probability matrix is denoted as $\boldsymbol{R} \to C^* \times R$. Given the pre-estimated cell counts within each spot, the cell allocation requirement for a specific cluster is determined by summing the estimated cells across all spots within that cluster. However, the FNN's allocation of single cells to ST clusters may lead to imbalances, resulting in predicted cell counts that deviate from the expected values. To address this issue, an iterative optimization strategy based on the probability matrix $\boldsymbol{R}$ is introduced, as illustrated in S9G Fig. This results in the creation of a list container $\mathcal{L}$:

$$\mathcal{L} = \left\{ \mathcal{L}_1, \ldots, \mathcal{L}_r, \ldots, \mathcal{L}_R \right\} \tag{17}$$

$\mathcal{L}_r$ is a set comprising the cells that pre-assigned to ST cluster $\mathcal{C}_r$.

(3) ***Assigning single cells to spots using linear assignment strategy***: Based on the cell counts within each spot, container $\mathcal{L}$, and the weighted likelihood similarity matrix $\boldsymbol{P}^*_{S' \times C^*}$, which is a subset of $\boldsymbol{P}'_{S \times C}$ and the spots in rows consisted only of highlighted spots in $\mathcal{H}$. The columns in $\boldsymbol{P}^*_{S' \times C^*}$ represent the cells source from cells $\boldsymbol{X}'$, with each column's values taken from $\boldsymbol{P}'_{S \times C}$ corresponding to the same cell names. To minimize the spread of mapped single cells from specific cell type to regions they do not belong to, $\boldsymbol{P}^*_{S' \times C}$ was adjusted as follows:

$$\boldsymbol{P}^*_{S' \times C^*} = \boldsymbol{P}^*_{S' \times C^*} - \left( 1 - \boldsymbol{H}'_{S' \times C^*} \right) \tag{18}$$

$\boldsymbol{H}'_{S' \times C^*}$ is a binary (0–1) matrix, where each column $\boldsymbol{H}'_{\cdot j^*}$ represents a binary vector of length $S'$. For each cell $j^*$ associated with cell type $j$, $\boldsymbol{H}'_{\cdot j^*}$ is defined as a subset of values extracted from the corresponding column $\boldsymbol{H}_{\cdot j}$ in the binary matrix $\boldsymbol{H}_{S \times K}$, retaining only the entries corresponding to the hotspots in $\boldsymbol{H}'$. The Jonker-Volgenant algorithm [23] was then employed to allocate single cells to specific spots to identify the ideal cell-to-spot assignment that maximize the following linear cost function (S9H Fig). For cluster $\mathcal{C}_r$:

$$\text{argmax} \sum_{t=1}^{\|\mathcal{C}_r\|_1} \sum_{j^*=1}^{\|\mathcal{L}_r\|_1} \left( \boldsymbol{P}^*_{(\mathcal{C}_r, \ \mathcal{L}_r)} \right)_{tj^*} \cdot \boldsymbol{A}_{tj^*} \tag{19}$$

where $\boldsymbol{P}^*_{(\mathcal{C}_r, \ \mathcal{L}_r)}$ is a subset of $\boldsymbol{P}^*_{S' \times C^*}$, with rows corresponding to spots from $\mathcal{C}_r$, columns represent cells from $\mathcal{L}_r$. $\boldsymbol{A}$ is a Boolean matrix, where $\boldsymbol{A}_{tj^*} = 1$ if single cell $j^*$ assigned to spot $t$. The constraint mentioned above guarantees that each cell is assigned only to one spot. The optimization is performed through the utilization of Python's *lapjv* package (version 1.3.24) [20,57]. These processes are iteratively applied to other clusters.

(4) ***Generating spatial coordinates for mapped single cells***: To represent SC coordinates within spatial slices, the minimum Euclidean distance between adjacent spots was calculated. Center coordinates for each spot were extracted, and random coordinates for assigned single cells were generated using the *spatstat.random::runifdisc* function, with the radius set to half of this minimum distance. This method established the spatial positions of all single cells. The mapping results were subsequently integrated into a "Seurat" object and provided to the user. Notably, for small-scale datasets, the FNN pre-assignment strategy can be omitted, with a linear assignment algorithm applied directly.

## 2. Benchmarking and simulations

**Simulation of spatial transcriptome data.** To assess the precision and robustness of Cell2Spatial, simulated ST datasets were generated using the Allen mouse brain SC dataset with brain ST data as the reference template. For

training, 50% of cells from each type were randomly selected, and gene expression perturbations (0%, 5%, 10%, 15%, and 20%) were introduced by shuffling expression values across cells to simulate noise. The top 2,000 HVGs were identified using the *FindVariableFeatures* function in Seurat, and PCCs were calculated to construct a cell–spot correlation matrix. Random spot-level cell counts were drawn from a Poisson distribution (*Lambda* = 5), and cells were assigned to spots according to the correlation matrix and sampled counts. The expression profile of each spot was then obtained by summing the expression values of its assigned single cells. For each simulated dataset, the number of cells, cell types, and spatial coordinates per spot were recorded for downstream evaluation. To further examine deconvolution performance and gene expression recovery, additional synthetic ST datasets were created with Poisson-distributed cell counts using *Lambda* values of 5, 10, 15, and 20.

**Benchmark metrics. *Accuracy index:*** Various spatial mapping tools (Cell2Spatial, CytoSPACE, CellTrek, Tangram, and Seurat) were used to project the remaining single cells of Allen mouse brain onto simulated ST sections with different levels of noise (Fig 2G). The performance of these tools was evaluated using the following formula.

$$AccuraySim_j = 2^{\frac{\sum_{t' \in \{t' :\ TN_{jt'} > 0\}} PN_{jt'}}{\sum_{t=1}^{S} PN_{jt}} \cdot \left(1 - RMSE(PN_{j\cdot},\ TN_{j\cdot})\right)} - 1 \tag{20}$$

$AccuraySim_j$[range: 0–1] represents the accuracy metric of cell type $j$ for effectively mapping to the simulated spots, which integrates both the relative abundance of cells in spatial spots and the RMSE between predicted and true counts. $TN_{jt}$ denotes the true count for cell type $j$ at spatial spot $t$, while $PN_{jt}$ represents the predicted count for the same type at spot $t$. $S$ refers to the total number of spots in the ST slice. $RMSE(PN_{j\cdot},\ TN_{j\cdot})$ is the RMSE between the predicted and true counts for cell type $j$ across all spots.

***Highly variable genes (HVGs) conservation:*** The HVGs conservation score is a critical metric for assessing the preservation of biological signals. To compute this score, Seurat's *FindVariableFeatures* function was utilized to identify HVGs in each ST dataset, both before and after mapping. For each batch, the target was to identify 500 HVGs. When evaluating the impact of the "group.size" parameter on mapping performance, the number of HVGs was set to 2,000. The HVG conservation score is defined as:

$$HVGs\ score = \frac{\left| HVGs^{synthetic} \cap HVGs^{mapped} \right|}{\min\left( \left| HVGs^{synthetic} \right|,\ \left\lceil HVGs^{mapped} \right\rceil \right)} \tag{21}$$

***Jaccard index:*** To evaluate the consistency between the mapped ST data and the synthetic ST data, the Jaccard index was used. The index, calculated at the cell type or SC level for each spot, is defined as:

$$Jaccard\ index = \frac{\left| spot^{synthetic} \cap spot^{mapped} \right|}{\left| spot^{synthetic} \cup spot^{mapped} \right|} \tag{22}$$

Here, $spot^{mapped}$ denotes the cell types or cells within the mapped spot, $spot^{synthetic}$ refers to those within the corresponding spot in the synthetic dataset.

***Pearson correlation coefficients (PCCs) and root mean square errors (RMSEs):*** Spatial deconvolution performance was compared using PCC and RMSE metrics. (1) For spatial mapping tools, cell type proportions in each spot were calculated based on the spatial distribution of assigned single cells. (2) For spatial deconvolution tools, proportions were inferred from the ST data using the corresponding SC dataset as a reference. Overall cell type composition in the ST slices was estimated by randomly selecting 2,000 spots, and the composition was computed as follows:

$$proportion = \frac{\sum_{t=1}^{2000} prop_{tj}}{\sum_{t=1}^{2000} \sum_{j=1}^{k} prop_{tj}}$$

(23)

where $k$ denotes the number of cell types, $t$ represents the index of spot. PCC and RMSE were then used to evaluate the concordance between synthetic and predicted compositions at the overall cellular composition level of the ST section. This procedure was repeated 100 times to generate error bars.

*Cosine similarity of gene expression:* Expression consistency between the mapped ST data and the synthetic ST data was evaluated by aggregating the profiles for each spot in the mapped ST data. This was done by summing the expression values of each gene across the single cells assigned to that spot, resulting in the recovered ST data. Consistency was then assessed by calculating the cosine similarity using 2,000 HVGs between the recovered and synthetic ST spots.

*Kullback–Leibler (KL) divergence:* To assess the global similarity of spatial structures, a *KL*-divergence-based approach from the CellTrek (version 0.0.94) package was utilized. The *CellTrek::SColoc* function computed a 2D grid kernel density for each cell type using *kde2d* from the "MASS" package (version 7.3–58.3), with a default bandwidth ($h = 100$) corresponding to the spatial distance between adjacent ST spots and a grid size ($n$) of 25. The *KL*-divergence was then calculated for the 2D density distributions between each pair of cell types from the mapped and synthetic data, with the synthetic data utilizing advanced SC information to achieve SC resolution.

### 3. Comparison of Cell2Spatial with publicly available spatial tools

**Cell2Spatial.** Cell2Spatial primarily employs default parameters for spot segmentation of both real and simulated 10× Visium ST data. Key parameters used include: normalize.method = '*SCTransform*', fix.cells.in.spot = TRUE, knn.spots = 5, and marker.selection = 'shannon'. For 10× Xenium, Visium HD, and Slide-seq V2 high-resolution ST data, the maximum cell count per spot was set to 1.

**CytoSPACE.** CytoSPACE [20] assigns individual cells to precise spatial spots by optimizing a correlation-based cost function, taking into account the estimated cell count per spot. This optimization is carried out using a shortest augmenting path algorithm. The "duplicated cells" option and "lapjv" solver were used during the CytoSPACE execution, and a standardized mean cell count of 5 per spot was maintained across all 10× Visium samples. For 10× Xenium, Visium HD, and Slide-seq V2 high-resolution ST data, the mean cell count was set to 2 for successful running.

**CellTrek.** CellTrek utilizes ST data to train a random forest model for predicting spatial coordinates, leveraging dimension reduction features shared with SC data [3]. The *traint* function from the "CellTrek" R package (version 0.0.94) was to obtain the co-embedding of ST and SC data for Visium samples. Subsequently, the *celltrek* function with default parameters ("reduction = 'pca', intp = T, intp_pnt = 10,000, intp_lin = F, nPCs = 30, ntree = 1,000, dist_thresh = 0.4, top_spot = 10, spot_n = 10, repel_r = 5, repel_iter = 10, keep_model = T) was used to project individual cells onto the ST coordinates. Cells were then assigned to their nearest spots based on the Euclidean distance. Notably, CellTrek failed to execute successfully for 10× Xenium, Visium HD, and Slide-seq V2 high-resolution data, thus it was excluded from the assessment and comparison of high-resolution ST data.

**Tangram.** Tangram [19] is a specialized deep-learning framework designed to reveal intricate spatial structures at the SC level. The "Seurat" object was converted into an "h5ad" file for all Visium samples, and Tangram was applied with default settings. The individual cells were mapped to ST spots by utilizing all accessible genes for each cell. To visualize these single cells on ST sections, random coordinates were generated for single cells based on their association with the respective spots.

**Seurat.** Seurat [18] employs a label transfer approach to map ST spots onto individual cell type, and considers each spot as a unit at the SC granularity level. The *FindTransferAnchors* function in Seurat was used in this study, with ST data as the query and the corresponding SC data as the reference. Subsequently, the *TransferData* function was utilized to assign SC labels (i.e., cell types) to the ST spots. Default parameters were used as specified in the official documentation.

**Cell2location.**  The guidelines on the Cell2location website (https://cell2location.readthedocs.io/en/latest/notebooks/cell2location_tutorial.html) were followed. The SC regression model was trained with the parameters "max_epochs = 100" and "lr = 0.002". The Cell2location model was then obtained using "max_epochs = 10,000".

**SpatialDWLS.**  The guidelines on the SpatialDWLS website (https://rubd.github.io/Giotto_site/articles/tut7_giotto_enrichment.html) were followed, with the parameter set to "n_cell = 20".

**RCTD.**  The guidelines on the RCTD GitHub repository (https://raw.githack.com/dmcable/spacexr/master/vignettes/spatial-transcriptomics.html) were followed, with the parameter set to "doublet_mode = full".

**Stereoscope.**  The guidelines on the website (https://docs.scvi-tools.org/en/stable/user_guide/models/stereoscope.html) were followed. The SC model was trained with parameters "max_epochs = 50" and the spatial model was trained with parameters "max_epochs = 100".

**DestVI.**  The guidelines on the DestVI website (https://docs.scvi-tools.org/en/stable/tutorials/notebooks/DestVI_tutorial.html) were followed. The SC model was trained with parameters "max_epochs = 50, lr = 0.001, number of training genes =2,000". The spatial model was trained with parameters "max_epochs = 100".

**SpaOTsc.**  The guidlines on the SpaOTsc GitHub repository (https://github.com/zcang/SpaOTsc) were followed. The spatial distribution of genes was obtained using the function 'issc.transport_plan' with parameters "alpha=0, rho=1.0, epsilon=0.1, scaling=False".

**novoSpaRc.**  The guidelines on the GitHub repository of novoSpaRc (https://github.com/rajewsky-lab/novosparc/blob/master/reconstruct_drosophila_embryo_tutorial.ipynb) were followed, with the parameters set to "alpha_linear = 0.5, loss_fun = square_loss, epsilon = $5 \times 10^{-3}$".

**SPOTlight.**  The guidelines on the SPOTlight GitHub repository (https://marcelosua.github.io/SPOTlight/) were followed, with the parameter set to "transf = uv, method = nsNMF".

**DSTG.**  The guidelines on the DSTG GitHub repository (https://github.com/Su-informatics-lab/DSTG) were followed.

**CARD.**  The guidelines on the CARD GitHub repository (https://yma-lab.github.io/CARD/documentation/04_CARD_Example.html) were followed, with the parameters set to "minCountGene = 100, minCountSpot = 5".

## 4.  Score-Guided Mapping Accuracy (SGMA)

Given the lack of information on precise cell location in real ST data, a strategy called SGMA was devised to gauge the performance of the mapping tools. Briefly, Seurat's *FindAllMarker* function (*logfc.threshold* = 0.25, *min.pct* = 0.25) was used to identify marker genes for each reference cell type within the corresponding SC dataset, and the top 20 overexpressed genes for each cell type were screened. The *AddModuleScore* function from Seurat was then applied in conjunction with the previously identified marker gene list to score the spots in the ST data. Thereafter, the scores of each cell type in spots were fitted to a Gaussian distribution, and the *p*-value for each score was calculated. Candidate spots potentially harboring a specific cell type were identified based on *p*-value ≤0.01 or 0.05. Finally, the performance of the mapping tool was quantified using the formula below:

$$SGMA\ index\ =\ \frac{1}{k} \cdot \sum_{j=1}^{k} \frac{\left| S'_j \cap S_j \right|}{\left| S'_j \cup S_j \right|}$$

(24)

$S'_j$ represents the set of spots that the mapping tool assigns to cell type *j*, $S_j$ represents the actual spots (determined by the Gaussian distribution mentioned above) where cell type *j* is located, $\left| S'_j \cap S_j \right|$ signifies the count of shared spots between $S'_j$ and $S_j$, $\left| S'_j \cup S_j \right|$ represents the total number of spots in the combined set of $S'_j$ and $S_j$, and *k* is the number of cell types. For a more intuitive comparison, accuracy metrics of various ST datasets were scaled to 0–1.

## 5. Cell counting in low-resolution spatial spots using DAPI from Visium fluorescence

Cell counts for the spots were obtained using the DAPI channel of the Visium fluorescence image, the spatial dataset was processed using "Squidpy" python package (version 1.6.0) [29]. The image was first smoothed using the smooth method, followed by segmentation with the watershed algorithm. Segmentation features were then calculated and integrated into the associated "AnnData" object. The cell counts for each spot were derived by extracting the "segmentation_label" information from the "AnnData" object.

## 6. Memory usage and time consumption analysis

To evaluate the memory usage and time consumption of Cell2Spatial, CytoSPACE, CellTrek, Tangram, and Seurat, 10 simulated ST datasets with 1,000–10,000 spots were generated using mouse brain data. The mapping programs were then executed with the corresponding SC data. Memory usage and time consumption were assessed using the "peakRAM" R package (version 1.0.2) [58].

## 7. Gene set enrichment analysis (GSEA)

GSEA was performed to determine whether single cells mapped onto thymic spatial sections by Cell2Spatial can provide insights into critical functional regions. Firstly, the single cells in the medullary region were compared with those in other areas using Seurat's *FindAllMarkers* function (Wilcoxon rank sum test). The functional gene sets were obtained from the C5 ontology gene sets within the MsigDB database. Subsequently, the "clusterProfiler" package (version 4.0.5) [44] and *bitr* function were used to convert gene symbols into Entrez IDs. The genes were sorted in descending order based on the fold change in expression. *GSEA* function was utilized with the "TERM2GENE = C5" parameter for enrichment analysis, and terms with adjusted *p*-values below 0.01 were considered significantly enriched. Finally, the *gseaplot2* function from the "enrichplot" package (version 1.18.4) was used to visualize the selected terms.

## 8. Spatial co-existence analysis of cell types in low-resolution spatial data using Cell2Spatial

The spatial co-existence of the different cell types was determined by projecting individual cells onto ST sections. Following the mapping of single cells to their respective spatial spots, the spatial co-localization among the various cell types was quantified using the formula as follows:

$$E_{ij} = \frac{|S_{ij}|}{min\left\{|S_i|, |S_j|\right\}}$$

(25)

$|S_i|$ and $|S_j|$ represent the number of spots containing cell types *i* and *j*, respectively, $|S_{ij}|$ denotes the number of spots containing both cell type *i* and *j*, and $E_{ij}$ is the coexistence index for cell types *i* and *j* (ranging from 0 to 1). $E_{ij} = 1$ signifies complete co-localization of the cell types in the ST sections, and $E_{ij} = 0$ indicates a lack of co-localization.

## 9. Statistical analysis

Pearson and Spearman correlation analysis, Student *t* test, Wilcoxon rank sum test, and Kruskal–Wallis test were performed as appropriate. *P*-value ≤0.05 was considered statistically significant. All statistical analyses were performed using R version 4.1.

## 10. Availability of datasets

The spatial data sources used in this study can be accessed through following links: (1) 10× Visium data of the mouse brain, available at [https://satijalab.org/seurat/articles/spatial_vignette.html], along with the corresponding SC atlas

[https://www.dropbox.com/s/cuowvm4vrf65pvq/allen_cortex.rds?dl=1] [59]; (2) 10× Visium data of the human thymus, accessible at [https://developmental.cellatlas.io/fetal-immune] [30], and the corresponding SC atlas of the thymus [DOI: https://doi.org/10.5281/zenodo.3572422] [31]; (3) 10× Visium data of the mouse kidney at https://www.10xgenomics.com/resources/datasets?query=&page=1&configure%5BhitsPerPage%5D=50&configure%5BmaxValuesPerFacet%5D=1000 and the GEO dataset GSE171406 [35], and the correspondence SC atlas (GSE129798) [11]; (4) 10× Visium data of the human DLPFC at [http://spatial.libd.org/spatialLIBD/] [37], and the matched SC atlas [https://cells.ucsc.edu/] [60]; (5) 10× Visium data of the human lung in GSE178361 [40], and the corresponding SC atlas [https://singlecell.broadinstitute.org/single_cell/study/SCP1219] [39]; (6) 10× Visium data of the human intestine in GSE158328 [9], and the SC atlas in GSE158702 [9]; (7) 10× Visium data of the human breast [https://doi.org/10.5281/zenodo.4739739], along with the SC atlas [GEO: GSE176078] [8]; (8) 10× Visium data of the renal cell carcinoma with TLS in GSE175540 [46], and the SC atlas [https://zenodo.org/records/4263972] [47]. For high-resolution ST data: (1) 10× Xenium dataset of the mouse brain is available from https://cf.10xgenomics.com/samples/xenium/1.0.2/Xenium_V1_FF_Mouse_Brain_Coronal_Subset_CTX_HP/Xenium_V1_FF_Mouse_Brain_Coronal_Subset_CTX_HP_outs.zip; (2) 10× Visium HD mouse brain dataset can be found at https://www.10xgenomics.com/datasets/visium-hd-cytassist-gene-expression-libraries-of-mouse-brain-he; (3) For the Slide-seq V2 platform, both the Mouse hippocampus dataset and the paired scRNA-seq reference were downloaded from https://satijalab.org/seurat/articles/spatial_vignette. Similarly, both the Mouse cerebellum dataset and the paired scRNA-seq reference were downloaded from the single cell portal project (https://singlecell.broadinstitute.org/single_cell/study/SCP948). Additionally, two mouse brain ST datasets with DAPI channels in fluorescence images for cell counting in spots were obtained from [https://support.10xgenomics.com/spatial-gene-expression/datasets/1.1.0/V1_Adult_Mouse_Brain_Coronal_Section_2] and [https://cf.10xgenomics.com/samples/spatial-exp/1.3.0/Visium_FFPE_Mouse_Brain_IF/Visium_FFPE_Mouse_Brain_IF_web_summary.html]. Finally, 32 simulated ST datasets with corresponding SC references were retrieved from https://drive.google.com/drive/folders/1pHmE9cg_tMcouV1LFJFtbyBJNp7oQo9J?usp=sharing. More details are provided in S1 Table.

## Supporting information

**S1 Fig. Spatial characterization and cell-type-specific genes in mouse brain tissue. (A) (i)** Clustering of spots from mouse brain spatial transcriptomics (ST) data, with clusters color-coded. **(ii)** Hematoxylin and eosin (H&E) staining image of mouse brain. **(iii)** Uniform Manifold Approximation and Projection (UMAP) showing clusters of spots inferred using Seurat [18] common processes. **(B)** UMAP plot showing the single-cell atlas of mouse brain. Each dot represents an individual cell. Cell types are marked by color codes. **(C)** Bubble plot showing the expression of five representative cell-type–specific marker genes selected from the top 30 overexpressed genes, as inferred by Cell2Spatial. Color intensity reflects the expression level of each gene in each cell type, while dot size corresponds to the proportion of cells expressing the gene. **(D–G)** Distribution of selected cell types with distinct spatial locations in mouse brain tissue, depicted using various mapping tools: (D) CytoSPACE, (E) CellTrek, (F) Tangram, and (G) Seurat. Each dot represents an individual cell. The data underlying this figure can be found at https://doi.org/10.5281/zenodo.17212677.
(TIF)

**S2 Fig. Synthetic data generation and performance evaluation for spatial transcriptomics (ST) mapping tools. (A)** Synthetic data generation strategy: This strategy comprises three main steps: **(i)** Randomly sampling cell counts for each spot based on a Poisson distribution, **(ii)** Assessing single-cell (SC) and spot similarity, and **(iii)** Aggregating the expression of nearby single cells based on the cell counts determined in step (i). To introduce variability, k% of genes are randomly perturbed with noise, resulting in synthetic ST data with diverse noise profiles (see "Materials and methods"). **(B)** Clustering of synthetic spots with varying noise levels on mouse brain tissue section. **(C)** Spatial heat maps showing the performance of publicly available mapping tools (CytoSPACE, CellTrek, Tangram, and Seurat) for aligning SC data

(with 5% added noise) to spatial spots in ST datasets simulated with five cells on average (see "Materials and methods"). To enhance clarity, we have presented only cell types with prominent spatial structures. The color intensity of each spot corresponds to the number of single cells assigned. **(D)** Strategy for estimating the number of cells in spots (see "Materials and methods"). **(E)** Concordance between the predicted and expected number of cells based on the synthetic mouse brain ST data, with dot size indicating the density of spots sharing the same predicted cell count. (i) *Lambda* = 5; (ii) *Lambda* = 20. Cell2Spatial (left panel); CytoSPACE (right panel). **(F)** Heat map showing the Pearson correlation coefficients (PCCs) between predicted and simulated benchmarking data for both Cell2Spatial and CytoSPACE. *Lambda* values ranging from 1 to 20 were used for random sampling the number of cells following a Poisson distribution. **(G)** Box and violin plots displaying the distributions of PCCs **(i)** and root mean square errors (RMSEs) **(ii)** between predicted and actual counts in spots for Cell2Spatial and CytoSPACE. *P*-values were determined using the Wilcoxon tests. **(H)** Bar plot showing the influence of the preset maximum cell count per spot on prediction accuracy in low-resolution ST data. (left) RMSEs quantifies the deviation between predicted and actual cell counts in the synthetic ST dataset, while (right) the PCCs assesses their concordance. The underlying data for this figure can be found at https://zenodo.org/records/17212677. (TIF)

**S3 Fig. Exploring the spatial architecture of the human thymus. (A) (i)** Clustering of spots from human thymus spatial transcriptomics (ST) data, with clusters color-coded. **(ii)** Hematoxylin and eosin (H&E) staining image of the thymus. **(iii)** Spatial location of each cluster on thymus ST section, highlighted by red color. **(B)** Uniform Manifold Approximation and Projection (UMAP) plot showing the single-cell atlas of human thymus. Each dot represents an individual cell, and cell types are marked by colors. **(C)** Spatial architectures of human thymus reconstructed using CytoSPACE **(i)**, CellTrek **(ii)**, Tangram **(iii)**, and Seurat **(iv)**. Each dot represents an individual cell. Cell types are marked by color codes. **(D)** Bar plot showing the number of cell types effectively mapped to spatial locations. "Reference" represents the total number of cell types in the single-cell atlas of the human thymus. The underlying data for this figure can be found at https://zenodo.org/records/17212677. (TIF)

**S4 Fig. Investigating the spatial architecture of the mouse kidney. (A)** Clustering spots of mouse kidney spatial transcriptomics (ST) data. Clusters are marked by color codes (left). Hematoxylin and eosin (H&E) staining of the corresponding kidney section (right). **(B)** Spots within each cluster on the mouse kidney ST section highlighted in red. **(C–F)** Mapping of epithelial cell transcriptomes from the mouse kidney single-cell atlas onto spatial spots of the corresponding ST section. Left: reconstructed spatial architectures with cells displayed using jitter within assigned spots. Right: the same representations with cells colored according to their known distance from the inner medulla. (C) CytoSPACE; (D) CellTrek; (E) Tangram; (F) Seurat. **(G)** Scatterplot showing the consistency between the cellular proportions in spatial architectures reconstructed with various mapping tools and cellular compositions predicted by the CARD spatial deconvolution tool [11]. The blue line denotes the linear fit, and the shaded area represents the 95% confidence interval. Different colors of points indicate distinct cell types. "*R*" represents the Pearson correlation coefficient (PCC). *P*-values were obtained using two-sided *t*-tests. **(H)** Bar plot showing the number of cell types effectively mapped to spatial locations. "Reference" represents the total number of cell types in the single-cell atlas of the mouse kidney. The underlying data for this figure can be found at https://zenodo.org/records/17212677. (TIF)

**S5 Fig. Exploring the spatial architecture of the human dorsolateral prefrontal cortex (DPLFC). (A)** Clustering spots of DPLFC spatial transcriptomics (ST) data. Clusters are marked by color codes (top). Hematoxylin and eosin (H&E) staining image of DLPFC tissue (bottom). **(B)** Uniform Manifold Approximation and Projection (UMAP) plot showing the single-cell atlas of human prefrontal cortex. Each dot represents an individual cell. Cell types are marked by color codes. **(C)** Spatial architectures of DLPFC tissue reconstructed using CellTrek **(i)**, Tangram **(ii)**, and Seurat **(iii)**, respectively.

Each dot represents an individual cell. Cell types are marked by color codes. **(D)** Heat map showing the distribution of cell types in anatomical structures of DLPFC. The intensity of the color indicates the cellular fraction within the specific anatomical region. CellTrek (left); Tangram (middle); Seurat (right). **(E)** Distribution of selected cell types in DLPFC tissue, reconstructed by different mapping tools. Each dot represents one cell. **(F)** Bar plot showing the number of cell types effectively mapped to spatial locations. "Reference" represents the total number of cell types in the single-cell atlas of the human prefrontal cortex. The underlying data for this figure can be found at https://zenodo.org/records/17212677. (TIF)

**S6 Fig. Spatial transcriptomics (ST) and single-cell atlases of multiple human tissues. (A) (i)** Clustering spots of human lung ST data. **(ii)** Hematoxylin and eosin (H&E) staining image of human lung tissue. **(iii)** Uniform Manifold Approximation and Projection (UMAP) showing the single-cell atlas of the human lung under COVID-19 state. **(B) (i)** Clustering spots of human intestinal ST data. **(ii)** H&E image of human intestinal tissue. **(iii)** UMAP plot showing the single-cell atlas of human intestine, annotated based on the marker genes provided by Fawkner-Corbett and colleagues [9]. **(iv)** Heat map showing pair-wise correlations of annotated cell types. **(C, D)** Clustering spots of human breast ST data (i); H&E staining image of human breast tissue (ii) [8]. (C) BRCA.1; (D) BRCA.2. **(E)** UMAP showing the single-cell atlas of human breast cancer patients [8]. Each dot represents an individual cell, and cell types are marked by color codes. The underlying data for this figure can be found at https://zenodo.org/records/17212677. (TIF)

**S7 Fig. Comparative analysis of spatial architectures and cellular proportions in Lung, Breast, and Intestinal tissues recovered by different mapping tools. (A–D)** Spatial architectures of human Lung (A), Breast (B and C), and Intestinal (D) tissues reconstructed by Cell2Spatial, CytoSPACE, CellTrek, Tangram, and Seurat. Each dot represents an individual cell and cell types are marked by color codes. **(E–H)** Scatter plots showing the consistency between the cellular proportions in spatial architectures reconstructed with various mapping tools and the cellular compositions predicted by CARD spatial deconvolution tool [11]. The blue line denotes the linear fit, and the shaded area represents the 95% confidence interval. Different colors of points indicate distinct cell types. "*R*" represents the Pearson correlation coefficient (PCC). *P*-values were obtained by two-sided *t*-tests. (E) Lung; (F) BRCA.1; (G) BRCA.2; (H) Intestinal. **(I)** Table summarizing the number of cell types effectively mapped to spatial locations by each tool. "Reference" represents the total number of cell types in the single-cell atlas of the human tissues. The underlying data for this figure can be found at https://zenodo.org/records/17212677. (TIF)

**S8 Fig. Characterization of cell distribution in mouse kidney and renal cell carcinoma (RCC) tissues using Cell2Spatial. (A)** Hematoxylin and eosin (H&E) staining image of mouse kidney tissue. **(B)** Spatial positioning of immune cell types in RCC tissue sections inferred by Cell2Spatial. The panel with the red border shows the corresponding H&E image of the RCC tissue. **(C)** Spatial plot showing the expression of selected marker genes in mouse kidney spatial transcriptomics (ST) data. T cell (left); Fibroblasts (Fb, middle); and Natural Killer (NK) cells (right). (TIF)

**S9 Fig. Assessment of multiple parameters for Cell2Spatial performance and assignment strategy. (A)** Boxplot showing the distribution of elapsed time for marker selection using Cell2Spatial's strategy compared with the Wilcoxon test implemented in the Seurat framework. Each point represents one dataset. **(B)** Boxplot comparing end-to-end mapping accuracy of Cell2Spatial with and without spatial weighting, using mouse brain Visium HD data (8 μm, down-sampled to 50,000 spots). *p*-value was calculated by the two-sided Wilcoxon test. **(C)** Bar plot summarizing the frequency with which different quantile cutoffs produced the highest overlap of highly variable genes (HVGs), indicating the most stable cutoff across conditions. To generate these results, Cell2Spatial was applied to map single cells onto

mouse kidney spatial transcriptomics (ST) data and reconstruct spatial expression profiles. HVGs were then computed from both reconstructed and original data across gene set sizes ranging from 50 to 3,000, and the consistency of HVG selection was quantified using the Jaccard index. This analysis was repeated across quantile cutoffs from 0.05 to 1 (step = 0.05). **(D)** Distribution of Getis-Ord G* indices for spatial spots corresponding to each cell type, derived from mouse kidney 10× Visium ST data. The black curve represents the observed Getis-Ord G* distribution, while the red curve shows the fitted normal distribution. **(E)** Scatter plots showing the predicted cellular compositions of spots by Cell2Spatial. The left panel displays results without smoothing using adjacent spots ($k=0$), while the right panel incorporates smoothing with the five nearest neighboring spots ($k=5$). True proportions are derived from a simulated ST dataset of the mouse brain. **(F)** Violin plots combined with boxplots showing the consistency between the estimated overall cellular compositions by Cell2Spatial and the simulated ground truth under different settings. The left panel shows root mean square error (RMSE) values, while the right panel shows Pearson correlation coefficients (PCCs). For each of 100 iterations, 1,000 spots were randomly sampled from the ST slice, and their aggregated cellular compositions were compared with the true proportions derived from the simulated dataset. **(G)** Schematic of the iterative strategy for preassigning single cells to ST clusters, utilizing a feedforward neural network (FNN) model. **(H)** Pseudocode illustrating the cell-to-spot assignment process. The underlying data for this figure can be found at https://zenodo.org/records/17212677.
(TIF)

**S1 Table. Summarize the data information utilized in this study.**
(XLSX)

**S2 Table. Median *k*-distances to the Astro cell type depicted by various mapping tools.**
(XLSX)

**S3 Table. Assessing the predictive performance of cell counts in spots (Cell2Spatial vs. CytoSPACE).**
(XLSX)

**S4 Table. Accuracy metrics of mapping tools based on synthetic datasets.**
(XLSX)

**S5 Table. Distance of T cell subsets at different developmental stages relative to the thymic medullary center.**
(XLSX)

**S6 Table. Performance metrics for mapping tools derived from mouse kidney data.**
(XLSX)

**S7 Table. Accuracy metrics of mapping tools on spatial transcriptomics (ST) data from multiple tissues.**
(XLSX)

**S8 Table. Co-existence atlas of cell types in mouse kidney.**
(XLSX)

**S9 Table. Co-existence atlas of immune cell types in tertiary lymphoid structures (TLSs) region of renal cell carcinoma (RCC) patients.**
(XLSX)

## Acknowledgments

We are grateful to Prof. Chuanle Xiao for providing valuable suggestions that helped improve this manuscript.

## Author contributions

**Conceptualization:** Liang Ma, Qing Xiong.

**Data curation:** Huamei Li, Jingchao Liu, Guige Wang, Zhenyu Liu.

**Formal analysis:** Huamei Li, Jingchao Liu.

**Funding acquisition:** Yiyao Liu, Liang Ma.

**Methodology:** Huamei Li.

**Project administration:** Lingyun Sun, Yiyao Liu, Qing Xiong.

**Software:** Huamei Li, Jingchao Liu.

**Supervision:** Lingyun Sun.

**Visualization:** Zhenyu Liu, Meng Cao, Cheng Peng.

**Writing – original draft:** Huamei Li, Guige Wang, Cheng Peng.

**Writing – review & editing:** Huamei Li, Cheng Peng, Yiyao Liu, Liang Ma.

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
