## [Editor Report · Decision Letter 0]

4 Jun 2025

Dear Dr Xiong,

Thank you for submitting your manuscript entitled "Cell2Spatial: precisely deciphering spatial transcriptomic spots at single-cell granularity" for consideration as a Methods and Resources Article by PLOS Biology. Please accept my sincere apologies for the delay in getting back to you as we consulted with an academic editor about your submission.

Your manuscript has now been evaluated by the PLOS Biology editorial staff, as well as by an academic editor with relevant expertise, and I am writing to let you know that we would like to send your submission out for external peer review.

Once your full submission is complete, your paper will undergo a series of checks in preparation for peer review. After your manuscript has passed the checks it will be sent out for review. To provide the metadata for your submission, please Login to Editorial Manager (https://www.editorialmanager.com/pbiology) within two working days, i.e. by Jun 06 2025 11:59PM.

Kind regards,

Richard

Richard Hodge, PhD

rhodge@plos.org

PLOS

---

## [Decision Letter · Decision Letter 1]

26 Aug 2025

Dear Qing,

Thank you for your patience while we considered your revised manuscript "Cell2Spatial: precisely deciphering spatial transcriptomic spots at single-cell granularity" for publication as a Methods and Resources article at PLOS Biology. Please accept my sincere apologies for the long delays that you have experienced during the peer review process. This revised version of your manuscript has been evaluated by the PLOS Biology editors, the Academic Editor and the three of the original reviewers at the previous journal.

Based on the reviews, I am pleased to say that we are likely to accept this manuscript for publication, provided you satisfactorily address the remaining points raised by the reviewers. In addition, please also make sure to address the following data and other policy-related requests that I have provided below (A-G):

(A) We routinely suggest changes to titles to ensure maximum accessibility for a broad, non-specialist readership. In this case, we would suggest a minor edit to the title, as follows. Please ensure you change both the manuscript file and the online submission system, as they need to match for final acceptance:

“Cell2Spatial is a computational framework that maps single cells to spatial transcriptomic spots to reconstruct tissue architecture”

(B) We note that your financial disclosure statement in the online submission form says that this study was not supported by any specific funding. Please provide additional details about any funding received to conduct the study, including the names of the funding agencies and grant numbers.

(C) You may be aware of the PLOS Data Policy, which requires that all data be made available without restriction: http://journals.plos.org/plosbiology/s/data-availability. For more information, please also see this editorial: http://dx.doi.org/10.1371/journal.pbio.1001797

-Supplementary files (e.g., excel). Please ensure that all data files are uploaded as 'Supporting Information' and are invariably referred to (in the manuscript, figure legends, and the Description field when uploading your files) using the following format verbatim: S1 Data, S2 Data, etc. Multiple panels of a single or even several figures can be included as multiple sheets in one excel file that is saved using exactly the following convention: S1_Data.xlsx (using an underscore).

-Deposition in a publicly available repository. Please also provide the accession code or a reviewer link so that we may view your data before publication.

Figure 2D-E, 2G-H, 3A-F, 3I-L, 4D, 4G, 4I, 5C, 5G, 6C, 6F-H, 7B, S1B-C, S2E-H, S3D, S4G-H, S5D, S5F, S6B, S7E-H, S9A-F

(D) Please also ensure that each of the relevant figure legends in your manuscript include information on *WHERE THE UNDERLYING DATA CAN BE FOUND*, and ensure your supplemental data file/s has a legend.

(E) Thank you for depositing the full analysis code in Github (https://github.com/lihuamei/Cell2Spatial.Reproduce). However, please note that we cannot accept sole deposition of code in GitHub, as this could be changed after publication. However, you can archive this version of your publicly available GitHub code to Zenodo. Once you do this, it will generate a DOI number, which you will need to provide in the Data Accessibility Statement (you are welcome to also provide the GitHub access information). See the process for doing this here: https://docs.github.com/en/repositories/archiving-a-github-repository/referencing-and-citing-content

(F) Please ensure that you are using best practice for statistical reporting and data presentation. These are our guidelines https://journals.plos.org/plosbiology/s/best-practices-in-research-reporting#loc-statistical-reporting and a useful resource on data presentation https://journals.plos.org/plosbiology/article?id=10.1371/journal.pbio.1002128

- If you are reporting experiments where n ≤ 5, please plot each individual data point.

(G) Please ensure that your Data Statement in the submission system accurately describes where your data can be found and is in final format, as it will be published as written there.

We expect to receive your revised manuscript within 1 month.

*Published Peer Review History*

*Press*

Best regards,

Richard

Richard Hodge, Ph.D.

rhodge@plos.org

Reviewer remarks:

Reviewer #1: The authors have successfully addressed all of my concerns. I have no other comments.

Reviewer #2: The revised manuscript is a significant improvement, and the authors have addressed my major concerns. I have some minor comments:

L23: "Several methods have been publicly disclosed. One common approach, like Seurat [18]"

Seurat is an R package with lots of methods. Can you be more specific?

L36: "In the case of low-resolution ST data (such as Spatial-based data)"

Isn't all ST spatial-based?

L68: "Spatial context is incorporated using Seurat's canonical correlation analysis (CCA) [18], which produces a Euclidean distance matrix from the Uniform Manifold Approximation and Projection (UMAP)"

Doesn't it use the PCA, not the UMAP?

"For high-resolution ST data, where spots typically capture single cells, it assigns a cell count of 1 per spot."

Both STOMICs and VisiumHD are sub-cellular resolution. Are the spots being binned to be approx. single cell sized? I would also be interested to know what the future development plans are for sub-cellular resolution datasets.

There are still some improvements that could be made when asserting claims about cell2spatial's performance compared to other tools in the results section of the manuscript. For instance the addition of a % performance difference/increase between tools, or quoting the Cell2Spatial metric and the range in the other tools (as the authors have done in the response to reviewers) could suffice.

L110, L128 (Only Jaccard index is quoted for Figure 2?), L153, L164-6 (how compares with the other tools?), L175-7 (other tools?), L194 (other tools?) etc.

L798: Dropbox link? Is there a better repository for this?

Reviewer #3: I am pleased to see that the authors have addressed most of the previous concerns. The additional analysis provided by the author strengthens the paper's claims. Overall, I have a positive view of the paper and support its publication. However, I suggest a few minor revisions are needed before the manuscript is finalized.

1. To address the previous comment from Reviewer 1. the authors have incorporated a comprehensive benchmarking study in their revised manuscript, citing Li et al., 2022. They have used their own simulated spatial transcriptomics (ST) datasets to evaluate the performance of Cell2Spatial and other tools. While this is a valuable exercise, the Li et al. paper provides its own set of benchmark datasets (already published). To further validate their claims of superior performance and provide a more direct comparison to the published literature, it would be highly informative to simply apply Cell2Spatial on those datasets and see how it performs on the publicly available benchmark datasets from the Li et al. study. I would recommend that the authors run their tool on these datasets and include the results in the manuscript.

2. Figure 2H, Consistency Result: The box plot shows the Jaccard index for single-cell mapping accuracy at different noise levels. Firstly, it is counterintuitive that for some tools, like CellTrek, the consistency seems to increase as the noise level increases. This could suggest that the consistency results are also subject to a high degree of randomness and using only one dataset is not enough for a consistent comparison. Additionally, the Jaccard index values for all tools are quite low, with the highest mean index being less than 0.20. The author should provide some comments/discussions on these phenomenon.

3. Figure 3H, Lower-Left Panel: The lower-left panel of Figure 3H, which displays the cell counts in a whole mouse brain ST slice using Cell2Spatial, shows a large contiguous area with the same cell count. The authors should provide an explanation for this phenomenon and exclude those spots from downstream analysis if appropriate.

4. Typo in Figure 1: In the upper-right portion of Figure 1, within the "Single-cell RNA-seq (SC) data" section, there is a typo in the legend. The "Cell Type" list shows "Type4" twice. This should be corrected for clarity.

I am confident that addressing these points will further enhance the quality of the manuscript and its contribution to the field.

---

## [Decision Letter · Decision Letter 2]

21 Oct 2025

Dear Qing,

On behalf of my colleagues and the Academic Editor, Selene Fernandez-Valverde, I am pleased to say that we can in principle accept your manuscript for publication, provided you address any remaining formatting and reporting issues. These will be detailed in an email you should receive within 2-3 business days from our colleagues in the journal operations team; no action is required from you until then. Please note that we will not be able to formally accept your manuscript and schedule it for publication until you have completed any requested changes.

IMPORTANT: We noted during our routine checks that the Data Availability Statement states that the structured spatial and corresponding single-cell data is deposited in a Google Drive (https://drive.google.com/file/d/13D9k1rb47XA7MBbpEcTgwxAlCXNEvdn-/view?usp=drive_link). Please note that we cannot accept deposition of data in a Google Drive and we ask that this data is deposited in the Zenodo database or similar. Due to the upcoming grant application, we have decided to move forward to editorial acceptance on your study, but I would be grateful if you could please move this data to an official data repository during the production process. Please update the Data Availability Statement in the online submission form and the manuscript file to reflect this.

PRESS

Best wishes, 

Richard

Richard Hodge, PhD

rhodge@plos.org

PLOS
